# Overexpression of Maize ZmC1 and ZmR Transcription Factors in Wheat Regulates Anthocyanin Biosynthesis in a Tissue-Specific Manner

**DOI:** 10.3390/ijms20225806

**Published:** 2019-11-19

**Authors:** Bisma Riaz, Haiqiang Chen, Jing Wang, Lipu Du, Ke Wang, Xingguo Ye

**Affiliations:** Institute of Crop Science, Chinese Academy of Agricultural Sciences, Beijing 100081, China; bisma.riaz75@gmail.com (B.R.); Chen_Haiqiang@yeah.net (H.C.); 13121260899@163.com (J.W.); dulipu@caas.cn (L.D.)

**Keywords:** wheat, MYB-type and bHLH transcription factors, genetic transformation, anthocyanin biosynthesis, gene expression

## Abstract

Maize *ZmC1* and *ZmR* transcription factors belong to the MYB-type and bHLH families, respectively, and control anthocyanin biosynthesis. In this study, *Agrobacterium*-mediated transformation was used to generate transgenic wheat plants that overexpress *ZmC1* and *ZmR* or both, with the objective of developing anthocyanin-enriched wheat germplasm. Three kinds of stable transgenic wheat lines were obtained. The integration of target genes in the transgenic wheat plants was confirmed by fluorescence in situ hybridization (FISH) analysis. We found that single overexpression of *ZmC1* regulates pigmentation in the vegetative tissues such as coleoptiles, auricles, and stems. The single overexpression of *ZmR* controls the coloration in reproductive tissue like spikelets and seeds. The simultaneous overexpression of *ZmC1* and *ZmR* showed the strongest pigmentation in almost all tissues. Furthermore, quantitative real-time PCR (qRT-PCR) analysis revealed that expression of the two transgenes, and of two conserved homologous and six associated structural genes involved in anthocyanin biosynthesis in wheat were greatly up-regulated in the transgenic plants. Similarly, quantitative analysis for anthocyanin amounts based on HPLC-MS also confirmed that the transgenic wheat plants with combined overexpression of *ZmC1* and *ZmR* accumulated the highest quantity of pigment products. Moreover, developing seeds overexpressing *ZmR* exposed to light conditions showed up-regulated transcript levels of anthocyanin biosynthesis-related genes compared to dark exposure, which suggests an important role of light in regulating anthocyanin biosynthesis. This study provides a foundation for breeding wheat materials with high anthocyanin accumulation and understanding the mechanism of anthocyanin biosynthesis in wheat.

## 1. Introduction

Anthocyanins are flavonoids responsible for a wide range of plant coloration, i.e., red to purple and blue-black colors found in leaves, flowers, fruits, pericarps, and seeds in diverse plant species [1]. These compounds play various protective roles against ultraviolet radiation, high or low temperature shock, salt stress, and attacks by microbial pathogens and insects [2,3]. Anthocyanin intake from plant-derived food in the human diet plays a protective role against coronary heart disease and also helps improve eyesight [4]. The antioxidant properties of anthocyanins also suggest their potential role in maintaining animal and human health [4,5,6]. Plant pigmentations at specific growth stages or in particular organs can be used as classical genetic markers because they are easily identified in the field. Particularly, anthocyanin pigmentation has been utilized as a reliable genetic marker in grasses [7,8]. Regulation of anthocyanin biosynthesis is intimately associated with environmental changes to enhance plant survival especially under stress conditions. Various factors, such as ultraviolet, visible light, cold, osmotic stress, nitrogen deficiency, and pathogen infection can induce anthocyanin biosynthesis [9,10]. A recent study provided evidence that the blue aleurone layer trait evolved as an environmental adaptation in rye, wheat, and barley, and light exposure can significantly increase the total anthocyanin content in barley grains [11].

The anthocyanin biosynthesis pathway has been elucidated in many plant species and the corresponding genes of this pathway have been isolated [12]. Several studies have revealed that the expression of some plant structural genes encoding anthocyanin biosynthesis enzymes (including chalcone synthase (CHS), chalcone isomerase (CHI), and flavanone 3-hydroxylase (F3H) dihydroflavonol 4-reductase (DFR), and anthocyanidin synthase (ANS) are controlled by a regulatory network consisting of R2R3-MYB, and basic helix-loop-helix (bHLH) transcription factors (TFs), and WD repeat (WDR or WD40) proteins [2,13,14,15]. It is well known that the regulation of anthocyanin accumulation in the vegetative and reproductive tissues of maize and other plant species is controlled by a ternary complex of these TFs called the MYB-bHLH-WD40 (MBW) complex [2,13,14,15,16]. The associated mechanism comprises the binding of these TFs onto structural or target genes and activating their expression in different plant species. These MYB and bHLH TFs are characterized by their specific spatial and temporal expression, and modulated by external environmental factors [17]. Moreover, their expression patterns are also influenced through hierarchical feedback regulation within the MBW complex [18,19]. Protein WD40 acts as a docking platform that allows the interaction between MYB and bHLH and stabilizes the TFs complex, without any intrinsic enzymatic function [20,21].

Currently, more than 100 R2R3-MYB proteins have been identified in maize [22] as controlling diverse cellular processes including cell metabolism, cell cycle, hormonal balance, and responses to external and internal environmental cues [22,23]. In maize, purple plant 1 (PL1) produces pigmentation in vegetative parts while colored aleurone 1 (C1) is responsible for colored grains [24]. MYB-type PL1 or C1 interacts with bHLH-type red (R1) or booster (B1) and the pale aleurone color 1 (PAC1), which is a WD40 protein, and results in an MBW complex formation, which ultimately regulates tissue-specific pigmentation at various plant developmental stages [25]. By using transposon tagging and sequencing cDNA strategies, *ZmC1* and *ZmR* genes involved in the regulation of anthocyanin biosynthesis in maize were cloned and characterized, respectively [26,27]. Ectopic expression of maize *ZmC1* and *ZmR* genes caused the accumulation of anthocyanins in root, stamen, and petal tissues in *Arabidopsis* and tobacco [28]. Similarly, transformation of maize *ZmC1* and *ZmR* genes in the tomato enhanced the flavonoid levels and produced anthocyanin in the leaves and fruit flesh [29]. Moreover, the transgenic expression of *ZmC1* and *ZmR-S* (seed-specific member of the *ZmR* gene family) in rice under an endospermic promoter produced different flavonoids in the endosperm of transgenic rice [30].

Wheat is the main cereal crop grown worldwide. Several wheat genes controlling anthocyanin accumulation in various parts of wheat have been documented. Specific homologous copies of the MYB and bHLH type genes were found to induce anthocyanin synthesis in wheat tissues, among which the bHLH gene *TaMyc1* (KJ747954) is a key regulator of anthocyanin biosynthesis in the pericarp [31]. Wheat *TaMpc1-A2* was involved in anthocyanin accumulation in wheat coleoptiles as a co-regulator of the MYB-type *Mpc1-1* genes [32]. Other anthocyanin biosynthesis-associated genes have also been investigated, including MYB type *purple anther* (*Pan*), *purple leaf blade* (*Plb*), and *red coleoptile* (*Rc*) along with the bHLH type *purple glume* (*Pg*) and *purple pericarp* (*Pp*) [33]. In a previous study, two wheat transcription factors encoded genes *TaPpm1* (*purple pericarp-MYB1*) and *TaPpb1* (*purple pericarp-bHLH1*) were characterized as activators of anthocyanin biosynthesis in purple pericarp. Mutations in the coding and promoter regions of these MYB or bHLH genes detected by sequencing analysis affected the activating ability of these TFs to the transcription of downstream structural genes [34]. Recently, the trigenic cluster *MbHF35* containing three genes (*HvMYB4H*, *HvMYC4H*, and *HvF35H*), which confers blue anthocyanin to grains, was identified in barley. *MbHF35* has only evolved within distinct *Triticeae* lineages, in which the three genes co-evolved via dispersed gene duplication [11].

*ZmC1* (MYB-type *C1*) and *ZmR* (bHLH *R*), which have strong expression in many tissues of some maize genotypes, have been transformed into other plants and regulated anthocyanin accumulation as expected in the transgenic plants [28,29,30]. Additionally, the exact mechanisms governing anthocyanin biosynthesis in wheat have not been fully revealed. In this study, two maize anthocyanin biosynthesis-related genes *ZmC1* and *ZmR* were transformed into wheat individually or in combination by *Agrobacterium*-mediated transformation to understand their regulatory function in anthocyanin biosynthesis in this plant species and to develop genetically modified purple wheat germplasm. The phenotype of different transgenic wheat plants and the expression levels of the two transgenes and some structural or regulatory network genes in the leaf tissues of different transgenic wheat plants were investigated. Total anthocyanin contents in the leaves and seeds of transgenic plants were determined and the effect of light on anthocyanin regulation was also evaluated.

## 2. Results

### 2.1. Purple Phenotype of Expressed R2R3-MYB and bHLH Type TFs in Immature Wheat Embryos and Derived Tissues after Transformation

To elucidate anthocyanin biosynthesis in wheat, three expression vectors (pWMB196 carrying maize transcription factor C1 encoded gene *ZmC1*, pWMB198 carrying maize transcription factor red/booster (R) encoded gene *ZmR*, and pWMB202 carrying *ZmC1* and *ZmR*), which regulate anthocyanin biosynthesis in maize, were constructed (Figure 1A–C). These vectors were transformed into immature wheat embryos through *Agrobacterium*-mediated transformation, in which pWMB202 was co-transformed with another expression vector, pWMB111, harboring a *Bar* gene selection marker (Figure 1D).

Five days after transformation on the rescue medium, the immature wheat embryos co-transformed with pWMB202 and pWMB111 in experiment HC-150 using non-purple wheat cultivar Jingdong18 that showed many purple parts. In contrast, the immature embryos transformed with pWMB196 in experiment NC-33 or pWMB198 in experiment NC-32, where both used non-purple wheat cultivar Fielder, and did not show any purple color (Figure 2A). One month afterward on the second round of P5 selection medium after transformation, purple pigmentation was clearly observed on the wheat callus transformed with pWMB202 in HC-150. Only a few purple spots were visualized under a microscope on the wheat callus transformed with pWMB196 in NC-33, and no purple spot was visualized on the wheat callus transformed with pWMB198 in NC-32 (Figure 2B). Subsequently, the regenerated shoots and roots from the transformed calluses with pWMB202 in HC-150 consistently displayed dark purple pigmentations due to anthocyanin accumulation (Figure 2C). In contrast, anthocyanin accumulation was not substantial in the transformed wheat tissues separately overexpressing *ZmC1* or *ZmR*.

### 2.2. Obtaining and Identifying Stable Transgenic Wheat Plants Expressing ZmC1 and/or ZmR

After regenerating the culture accompanying phosphinothricin (PPT) selection for 2.5 months (about 76 days), the regenerated plantlets with well-developed root systems were transferred into pots for further growth in a growth chamber. During the tilling stage, PCR was performed to detect the putative transgenic plants after genomic DNA extraction. Five out of 11 independent putative transgenic plants were confirmed to be positive for both transgenes *ZmC1* and *ZmR* derived from HC-150 (Appendix A), and showed purple pigmentation. Among these five independent putative transgenic plants, only two plants survived and produced seeds. All 16 independent putative transgenic plants were confirmed to be positive for *ZmC1* derived from experiment NC-33 (Appendix A) and showed purple pigmentation only on leaf auricles. Two independent putative transgenic plants were confirmed to be positive for *ZmR* derived from experiment NC-32 (Appendix A) and did not show any clear pigmentation.

Up to the T4 generation, 12 plants were randomly picked from each transgenic line and detected by PCR for the corresponding transformed transgene(s). Lastly, three types of stable transgenic lines (AL-30, AL-31, and AL-32 with *ZmC1* from Fielder, AL-40 and AL-41 with *ZmR* from Fielder, and AL-44 and AL-45 with both *ZmC1* and *ZmR* from Jingdong18) were obtained. Furthermore, the integration of *ZmC1* and *ZmR* in the stable transgenic wheat lines was detected through fluorescence in situ hybridization (FISH) analysis using the DNA of plasmids pWMB196 and/or pWMB198 as probes. A pair of chromosomes showed strong green hybridization signals on their telomeric or sub-telomeric regions in the three stable transgenic lines AL-30, AL-40, and AL-45 (Figure 3). The FISH results demonstrated that the transgenes *ZmC1* and *ZmR* were integrated into the genome of transgenic lines AL-30 and AL-40, respectively, and both genes were integrated into the genome of transgenic line AL-45. The three transgenic lines were clearly stable.

### 2.3. Phenotype of Three Types of Stable Trsansgenic Lines Carrying ZmC1 and/or ZmR Transgenes

Transgenic lines AL-30 and AL-31 expressing *ZmC1* and AL-40 and AL-41 expressing *ZmR* showed tissue-specific anthocyanin accumulation (Figure 4 and Figure 5). In detail, transgenic lines AL-30 and AL-31 showed very dark pigmentation in the vegetative tissues like coleoptiles, seedlings, auricles, and stems (Figure 4 and Figure 5, Appendix A). Transgenic lines AL-40 and AL-41 accumulated the pigments only in the reproductive parts of the plants such as glumes and seeds (Figure 4 and Figure 5, Appendix A). Under the same growing conditions, transgenic lines AL-45 and AL-44 harboring *ZmC1* and *ZmR* genes showed a high level of pigment accumulation in almost all the tissues including seedlings, leaves, auricles, stems, spikes, immature seeds, and dry seeds (Figure 4B–E and Figure 5A–C, Appendix A).

### 2.4. Expression Profiling of the Two Transgenes ZmC1 and ZmR as Well as Their Orthologous Genes in Wheat in the Three Types of Transgenic Lines

The two corresponding native genes *TaPpm1* (GenBank accession number: MG066451) and *TaPpb1* (GenBank accession number: MG06645) of the two foreign transferred genes *ZmC1* and *ZmR,* which are identical to the *TaMpc1* and *TaMyc1* genes, respectively, are involved in the flavonoid and anthocyanin biosynthesis pathways in wheat. Quantitative real-time PCR (qRT-PCR) was performed to evaluate the comparative transcript levels of the four genes *ZmC1*, *ZmR*, *TaPpm1*, and *TaPpb1* in the leaves of the three types of stable transgenic lines (AL-45 and AL-44 containing *ZmC1* and *ZmR*, AL-30, AL-31, and AL-32 containing *ZmC1*, and AL-40 and AL-41 containing *ZmR*) as well as in the wild-type plants from Fielder and Jingdong18. Results indicated that *ZmC1* and *ZmR* were highly expressed in the three types of transgenic lines, especially *ZmC1* in the transgenic lines AL-30, AL-31, AL-32, AL-45, and AL-44, and the *ZmR* in the transgenic lines AL-45, AL-44, AL-41, and AL-42 (Figure 6A). Using AL-45 as a control, the relative expression level of *ZmC1* was significantly enhanced (elevated by three-fold to four-fold) in the *ZmC1* containing transgenic lines AL-30, AL-31, and AL-32. However, no pigmentation was observed in the leaves of these transgenic lines. The relative expression level of *ZmR* in the transgenic lines AL-45 and AL-44 containing both transgenes was higher that in other transgenic lines only containing *ZmR* (Figure 6A). Additionally, the expression level of *TaPpb1* was extensively enhanced in the transgenic lines AL-45 and AL-44. The expression level of *TaPpm1* was also clearly elevated in all three types of transgenic lines, compared with that in the wild type (Figure 6B). These results suggest that the overexpression of the exogenous genes *ZmC1* and *ZmR* up-regulates the expression of the endogenous orthologous genes *TaPpm1* and *TaPpb1*.

### 2.5. Expression Profiling of Wheat Native Anthocyanin Biosynthesis-Related Genes in the Three Types of Transgenic Lines

A selection of six wheat endogenous structural genes encoding the enzymes involved in anthocyanin pathways including *TaCHS* (unigene c54121_g1_i2), *TaCHI* (unigene c49033_g1_i1), *TaF3H* (unigene c57117_g3_i1), *TaF3′H* (unigene c55981_g1_i1), *TaDFR* (unigene c58412_g2_i1), and *TaANS* (unigene c57117_g3_i5) (Appendix A) was performed to investigate the possible regulation mechanism in the transgenic wheat lines by the two target genes *ZmR* and *ZmC1*. Their expression levels were evaluated in the leaves of the three types of stable transgenic lines as well as both wild-type plants. By taking the wild-type Fielder as a reference sample, we found that the relative expression levels of the six structural genes were all significantly up-regulated in the transgenic lines AL-45 and AL-44 when compared with the other two types of transgenic lines and the two wild types (Figure 7). Moreover, the relative expression levels of *TaCH*S, *TaCHI*, *TaF3H*, *TaF3′H*, and *TaANS* were significantly increased (five-fold to ten-fold) in the *ZmC1* carrying transgenic lines (AL-30, AL-31, and AL-32), and *ZmR*-containing transgenic lines (AL-40 and AL-41), in contrast to the wild types. Nevertheless, there were no big differences in expression levels of the six native genes between *ZmC1* transgenic lines (AL-30, AL-31, and AL-32), and *ZmR* transgenic lines (AL-40 and AL-41). However, some of the six native genes showed higher expression in the transgenic lines than in the wild type. However, the transcript levels of *TaDFR* were not significantly upregulated in *ZmR-*containing transgenic lines (AL-40 and AL-41) (Figure 7).

### 2.6. Exposure To Strong Light Promoted Anthocyanin Biosynthesis in the Developing Seeds of ZmR Stable Transgenic Lines

To elucidate the influence of light on the production of anthocyanins, the developing seeds of the two stable transgenic lines AL-40 and AL-41 derived from the transformation of pWMB198 containing *ZmR* gene were exposed to strong light under controlled field conditions. From top to bottom in a spike, the developing seeds in one side were exposed to light by removing their lemma, and the ones in another side were not exposed to light (closely protected by lemma) and used as a control. Six days after the treatment, all seeds exposed to light were found to accumulate much more anthocyanin pigment in comparison with the seeds without light exposure in the same spike (Figure 8), which indicates that light increases anthocyanin accumulation.

To clarify if light affects the expression levels of *ZmR* and other wheat native genes involved in anthocyanin biosynthesis, light exposed seeds and control seeds of the transgenic line AL-40 were analyzed by qRT-PCR to evaluate the comparative transcript levels of *ZmR* and seven wheat regulatory and structural genes *TaPpb1*, *TaCHS*, *TaCHI*, *TaF3H*, *TaF3′H*, *TaDFR*, and *TaANS*. In this experiment, the untreated AL-40 seeds were used as the reference sample to calculate the relative expression levels. Transcript levels of the two regulatory (*ZmR* and *TaPpb1*) and six wheat structural genes (*TaCHS*, *TaCHI*, *TaF3H*, *TaF3′H*, *TaDFR*, and *TaANS*) were all significantly upregulated (elevated by two-fold to seven-fold) in the light exposed seeds as compared to the control seeds in the same spike in lines AL-40 and AL-41 (Figure 9). This result suggested that light exposure enhanced the expression levels of related genes to anthocyanin biosynthesis and boosted anthocyanin accumulation in the developing seeds of *ZmR*-containing transgenic wheat plants.

### 2.7. Determination of Anthocyanin Content in the Three Types of Transgenic Wheat Plants

Total anthocyanin content was determined in the fresh leaves and dry kernels of the three types of transgenic wheat plants and their wild-type plants through HPLC-MS, which were measured in microgram per gram of fresh and dry weight of the sample, respectively. The content of anthocyanin such as cyanidin 3-O-glucoside, pelargonidin 3-O-glucoside, peonidin 3-O-glucoside, petunidin 3-O-glucoside, and peonidin 3-O-hexoside were detected, and only the content of cyanidin 3-O-glucoside had very significant differences between the green leaves of the wild type leaves and the purple leaves of transgenic lines. Cyanidin 3-O-glucoside had the highest proportion of anthocyanin content and presented as total anthocyanin levels. The highest anthocyanin contents occurred in transgenic lines AL-44 and AL-45 containing both *ZmC1* and *ZmR*, which were 96.52 ± 3.33 µg/g and 102.2 ± 8.03 µg/g of fresh weight in the leaves, and 10.18 ± 0.18 µg/g and 12.95 ± 0.84 µg/g of dry weight in the grains (Appendix A), respectively. Comparison analysis indicated that anthocyanin content of the other two types of transgenic lines was not significantly increased when compared to the wild-type plants (Figure 10, Appendix A).

## 3. Discussion

Research in model plants and maize revealed that anthocyanins are synthesized through a branch of the flavonoid biosynthesis pathway [10,35,36]. The genes involved in this pathway are categorized in two different groups: structural genes and regulatory genes. Structural genes encode the enzymes that catalyze the different reaction steps in the pathway and regulatory genes encode TFs that interact with the structural genes at their promoter regions, which allow precise temporal and spatial coordination of their transcription. This results in the production of anthocyanins [37]. Recent models indicate that R2R3 MYB-type and bHLH-type TFs physically interact with another protein WD40 repeat (WDR) and form a MYB-bHLH-WDR (MBW) transcriptional complex and regulate the transcription activation of the structural genes [20,37,38,39].

To understand anthocyanin biosynthesis in wheat in this study, two maize regulatory genes *ZmC1* and *ZmR* were overexpressed in wheat, and three kinds of stable transgenic lines were obtained, containing *ZmC1* (lines AL-30, AL-31 and AL-32), *ZmR* (lines AL-40 and AL-41), and *ZmC1* + *ZmR* (lines AL-45 and AL-44). Consequently, the stable transgenic wheat lines were identified by FISH and the integration of *ZmC1* and *ZmR* genes at the telomeric or sub-telomeric regions of wheat chromosomes was consistent with recently published results for the chromosomal integration of *GUS* and *Bar* genes in transgenic wheat plants [40]. Overall, the transgenic lines AL-45 and AL-44 showed an enhanced level of anthocyanin accumulation (Figure 4, Appendix A), which suggests that the complex formation increased anthocyanin deposition. The higher anthocyanin accumulation level in the transgenic wheat tissues overexpressing *ZmC1* and *ZmR* under the ambient growth conditions also suggested the regulation of anthocyanin biosynthesis through the MYB-bHLH-WD40 (MBW) complex in this plant species.

Moreover, anthocyanin biosynthesis in different tissues is often regulated by spatial expression of MYB or bHLH proteins [39,40,41]. Phenotypic characterization of the transgenic wheat lines reflected the tissue-specific expression of MYB and bHLH-type TFs: MYB-type *ZmC1* regulates the accumulation of pigments only in the vegetative tissues while bHLH-type *ZmR* controls the pigmentation only in the reproductive tissues (Figure 4 and Figure 5). In the present study, transgenic lines AL-30 and AL-31 with *ZmC1* transgene accumulated pigmentation only in the vegetative tissues like coleoptiles, seedlings, auricles, stems, and roots (Figure 4, Appendix A), which indicated the tissue-specific expression of R2R3-MYB type TF in wheat by interacting with downstream structural genes. This finding is consistent with previous studies in which wheat MYB TF controls the pigmentation in vegetative tissues like coleoptiles [38,39,40,41,42].

Additionally, we observed that *ZmC1* induced pigmentation in the wheat calluses during the selection regime after transformation, which is similar to the previous report in wheat, where MYB-type TF (*TaPpm1*) alone produced pigmentation in wheat callus [35]. MYB and bHLH TFs regulate tissue-specific anthocyanin accumulation, but the tissue specificity of each TF for anthocyanin biosynthesis varies among plant species [2,42,43]. Recent studies showed that the expressions of *ZmC1* and *TaPpm1* (which was identical to *TaMpc1*) in wheat control the pigmentation in vegetative tissues. However, *ZmC1* in maize and *OsC1* in rice regulated anthocyanin biosynthesis in reproductive tissues like seeds [2,42,43]. Thus, the phenotype of the transgenic lines AL-30 and AL-31 revealed the unique aspect of *R2R3*-*MYB* type TF in wheat, which functioned as a vegetative tissue-specific regulator of anthocyanin pigmentation in wheat, in contrast to the grain color regulation in maize and rice [2,43].

In previous studies, bHLH TF (termed *TaPpb1* in the present study) was proposed as a candidate gene and specifically participated in anthocyanin synthesis in wheat grains [35,36]. Likewise, in our study, transgenic lines AL-40 and AL-41 showed anthocyanin accumulation only in the reproductive tissues like glumes and seeds (Figure 5, Appendix A), which revealed that bHLH TF regulated the anthocyanin biosynthesis in a tissue-specific manner in the reproductive tissues of wheat. A recent study elucidated that *TaPpb1*, which is identical to *TaMyc1*, is a seed-specific expression gene. This was uncovered by investigating the roles of five seed-specific *cis-*elements in the promoter region of *TaPpb1* [36].

Like the *C1* gene, tissue specificity of *R* gene expression also varied among different plant species. In a recent study, a homologous bHLH *AetMYC* on wheat chromosome 2D regulated anthocyanin production in the coleoptile of *Aegilos tauschii*, which is the ancestor of the wheat D genome [44]. Similar to maize, the *b* alleles *B-peru* and *B-1* modulated the specific anthocyanin accumulation in seeds and vegetative tissues, respectively [45]. Unlike the other species, we conclude that *C1* and *R* have different spatial expression in wheat based on the phenotypes of transgenic lines AL-30 and AL-40 (Figure 4 and Figure 5).

An integrated model for the gene regulation network determining anthocyanin biosynthesis has been described in previous studies [46,47]. In the present study, expression profiling analysis was performed on the leaf tissues of the three types of transgenic lines for maize and wheat MYB and bHLH TFs, and wheat native structural genes involved in anthocyanin synthesis. A study in *Medicago truncatula* found that the bHLH TF MtTT8 regulated its own expression as part of a positive feedback loop, through the MBW complex formation, which participates in the regulation of anthocyanin biosynthesis [19]. This is consistent with our result for *ZmR* expression profiling, in which the transgenic lines AL-45 and AL-44 with both TFs showed the highest transcript level of *ZmR* (Figure 6). However, *ZmC1* showed low transcript levels in the transgenic lines AL-45 and AL-44 as compared to those in lines AL-30, AL-31, and AL-32 (Figure 6).

In *Arabidopsis*, the genes encoding enzymes of the anthocyanin biosynthesis are grouped into two classes: early biosynthesis genes (EBG*s*), including CHS, CHI, F3H, and F3′H involved in flavonoid pathways, and late biosynthesis genes (LBGs) like DFR and anthocyanin acyltransferase (AAT) catalytic enzymes for anthocyanin production [10]. EBGs are regulated by R2R*3*-MYB type TFs, while LBGs are transcriptionally activated through the MBW complex [48]. In maize, both EBGs and LBGs are regulated through ternary complex MBW consisting of MYB-type *ZmC1*, bHLH-type *ZmR*, and WD40 protein PAC [2]. In the present study, expression profiling of the corresponding genes in wheat also showed the same trend as wheat EBGs and LBGs, which were highly expressed in the transgenic lines AL-45 and AL-44 due to the MBW complex formation (Figure 7). In contrast, EBG expression was a little higher in the *ZmC1* lines (AL-30, AL-31, and AL-32) than the *ZmR* lines (AL-40 and AL-41) except *TaCHI* (Figure 7). However, LBGs were not highly expressed in the transgenic lines AL-30, AL-31, and AL-32 containing *ZmC1*, or AL-40 and AL-41 containing *ZmR* as compared to the transgenic lines AL-45 and AL-44 carrying both transgenes. This is due to the absence of the MBW complex, which suggests that activation of LBGs in wheat showed the same trend as that in maize and *Arabidopsis* in leaf tissues [2,48].

Since anthocyanin biosynthesis is a metabolically expensive process, it is closely regulated with respect to surrounding environmental conditions. Thus, the regulation of anthocyanin biosynthetic genes is associated with many environmentally responsive signaling pathways. In the present study, qRT-PCR confirmed high transcript levels of anthocyanin-related genes in the developing seeds exposed to light as compared to the control seeds of the *ZmR* transgenic line (Figure 9), which suggests that anthocyanin regulation is affected by light. Consistent with previous studies [49,50], light can induce the expression of related genes in the anthocyanin biosynthesis pathway. For example, a recent study also found that light exposure significantly regulated the expression of *MbHF35* and increased the blue anthocyanin content in barley seeds [11]. This result suggests that corresponding or structural genes in wheat seeds accumulatively affect the expression of the *ZmR* gene under light conditions. Namely, light might promote the expression of some seed-specific genes and then promote the expression of *ZmR* and anthocyanin biosynthesis in grains of the transgenic wheat line. The exact reasons for this phenomenon need to be investigated further.

According to previous studies, several derivatives account for the total anthocyanin contents in cereals, with cyanidin 3-O-glucoside as the main derivative found in cereal grains [51,52]. Quantitative analysis of anthocyanin contents in the proposed study also indicated that cyanidin 3-O-glucoside is the major anthocyanin form present in the leaf and seed tissues of the transgenic wheat plants containing maize *ZmC* and/or *ZmR* genes. Transgenic lines AL-44 and AL-45 had the highest anthocyanin contents in their leaves and seeds as expected (Figure 10). Our results confirmed that simultaneous overexpression of both maize genes in wheat produced high anthocyanin pigmentation due to the MBW complex formation.

## 4. Materials and Methods

### 4.1. Plant Material and Growth Conditions

Two common wheat (*Triticum aestivum* L., AABBDD, 2n = 42) genotypes Fielder (spring type) and Jingdong18 (winter type) were used in this study. Fielder was directly sown in a growth chamber. Jingdong18 seeds were first germinated in culture dishes (90 mm in diameter and 20 mm in height) at room temperature (25 °C) for two days and then exposed to 4 °C for 30 days for vernalization before sowing. Dry wheat seeds or germinated seedlings were planted in pots (20 cm in diameter and 30 cm in height) filled with substrate peat moss (Parnumaa, Estonia) and mixed patterns of fertilizer release (Osmocote Extract, Heerlen, Netherlands). Optimal growth conditions including a temperature of 24 °C, a photoperiod regime of 16 h light/8 h darkness, light intensity of 300 mol/m2/s, and humidity of 45% were maintained. Water was supplied once a week at subsequent growth stages. Aphids were limited by using sticky colored cards (Zhengzhou Oukeqi Instruments Ltd., Zhengzhou, China). Powdery mildew was controlled by application of triadimefon (Jinan Luba Pesticides Co., Jinan, China).

### 4.2. Construction of Expression Vectors Containing Transcriptional Factors Involved in Anthocyanin Biosynthesis

Full sequences of maize regulatory genes *ZmC1* and *ZmR* involved in anthocyanin biosynthesis were amplified separately from the plasmid pWMB022 (Appendix A), in which *ZmC1* and *ZmR* were cloned from maize based on previous publications [26,27]. The amplified *ZmC1* fragment was double digested with *Sma*I and *Spe*I. The plasmid of another vector pWMB006 (Appendix A) [53] was also digested with the same enzymes. The digested products were ligated together to form a new vector named pWMB195. Next, pWMB195 was digested with *Hin*dIII and the large fragment (3.8 kb) was harvested. At the same time, another vector, pWMB111, (Figure 1D, Appendix A, data unpublished) containing the *Bar* selection gene was also digested with *Hin*dIII. Later, the harvested 3.8-kb fragment was inserted into the digested pWMB111 by ligation to form the final expression vector pWMB196 carrying *ZmC1* linked with the *Bar* gene in its T-DNA region (Figure 1A). Similarly, the amplified *ZmR* fragment and plasmid pWMB006 were digested with *Kpn*I and *Spe*I, and then both digested DNA products were ligated to form a new vector pWMB197. Subsequently, pWMB197 and pWMB111 were digested with *Hind*III, and the larger fragment of 4.8 kb in length was harvested and ligated with pWMB111 to form the final expression vector pWMB198 containing ZmR linked with the *Bar* gene in its T-DNA region (Figure 1B).

To construct the expression vector containing *ZmC1* and *ZmR* genes linked on the same T-DNA region, plasmids pWMB195 carrying *ZmC1* and pWMB190 were double digested with *Bam*HI and *Sac*I, and then ligated together to incorporate the *ZmC1* gene into pWMB190. Furthermore, the ligated product pWMB190-ZmC1 and pWMB197 were digested with *Hin*dIII, and ligated to each other to form the expression vector pWMB202 harboring both target genes *ZmC1* and *ZmR* (Figure 1C), which was co-transformed with the vector pWMB111 harboring the *Bar* gene as a selection marker for wheat transformation. Lastly, the four vectors pWMB196, pWMB198, pWMB202, and pWMB111 were transformed into the *Agrobacterium* strain C58C1, using the tri-parent hybridization method [54] for wheat transformation.

### 4.3. Agrobacterium-Mediated Transformation Using Immature Wheat Embryos

Wheat plants sown in the growth chamber were tagged at the anthesis stage and their heads were harvested 14 days post anthesis (DPA). Immature wheat grains were carefully collected. Surface sterilization was done in aseptic conditions using 70% ethanol for 1 min and 5% sodium hypochlorite (NaClO) for 15 min, which was followed by five rinses with sterile water. *Agrobacterium*-mediated transformation was performed using the exclusive method developed by Japan Tobacco Company [55] with minor modifications. The transformation was performed as follows. Immature embryos were carefully dissected from the sterilized grains under a stereoscopic microscope, incubated with *Agrobacterium* strain C58C1 harboring expression vectors pWMB196, pWMB198, and pWMB202 or pWMB111, for transformation or co-transformation for 5 min in WLS-inf medium at room temperature. The transformed immature embryos were co-cultivated on WLS-AS medium for 2 days at 25 °C in the dark with the scutellum facing upward. Then, the embryonic axes were removed with a scalpel and the scutella were shifted onto WLS-Res medium. After five days, the scutella were moved onto WLS-P5 medium for callus induction. After 14 days, the calluses were divided into two slices and consistently shifted on WLS-P10 medium in the dark for three weeks. The developed embryonic calluses were then allowed to differentiate on LSZ-P5 medium at 25 °C and light intensity of 100 µmoles/m^2^/s. Subsequently, the regenerated shoots were transferred into sterile cups containing MSF-P5 medium for elongation and root formation. When the plantlets had reached the height of 10 cm around with developed roots, they were transplanted into pots and cultured in a growth chamber maintained at 25 °C with a light intensity of 300 µmol/m^2^/s.

### 4.4. DNA Extraction and PCR Amplification

Genomic DNA was extracted from the young leaves of transgenic and wild-type plants using the NuClean Plant Genomic DNA Kit (CW Bio Inc., Beijing, China). The DNA pellet was dissolved in sterile distilled water and diluted to 200 μg μL^−1^ for use as a template for PCR amplification. PCR was carried out in 15.0 µL reactions containing 6.0 µL distilled water, 1.0 µL template DNA (100 ng µL^−1^), 0.5 µL of each primer (10 µmol L^−1^), and 7.5 µL 2 × Taq MasterMix (containing Mg^2+^ and dNTPs, CW Bio Inc., Beijing, China). PCR amplification was performed in a Biometra Professional Thermal Cycler (Göttingen, Germany) with an initial denaturation at 95 °C for 3 min, 35 cycles of 30 s at 95 °C, 30 s at 60 °C, 1 min at 72 °C, and a final extension of 5 min at 72 °C. PCR products were separated on 1% agarose gels and visualized by UV light. PCR primer sequences used in this study for detecting target genes are given in Appendix A.

### 4.5. Chromosome Preparation and Fluorescence In Situ Hybridization

Wheat seeds were germinated on porous paper at room temperature, and their rapidly growing roots were collected. The root tips were treated with nitrous oxide (NO) for 2 h and then fixed with 90% glacial acetic acid for 5 min. The chromosome preparation was accomplished using a previously established method [56]. Fluorescence in situ hybridization (FISH) was performed to produce the stable transgenic wheat lines AL-30 (*ZmC1*), AL-40 (*ZmR)*, and AL-45 (*ZmC1 + ZmR*) lines in the T_3_ generation using vectors pWMB196 containing *ZmC1* and pWMB198 containing *ZmR* as probes, which were labelled with fluorescein−12-dUTP. Hybridization procedures were performed, and the prepared slides were visualized using an Olympus BX-51 microscope with a Photometric SenSys Olympus DP70 CCD camera [56].

### 4.6. RNA Extraction and Quantitative Real-Time PCR Assay

Total RNA was extracted from 50 mg leaf samples of transgenic wheat plants using TRIzol reagent (Invitrogen, Carlsbad, California, USA), according to the manufacturer’s instructions with minor modifications. Complementary DNA (cDNA) was synthesized by a reverse transcription reaction using the PrimeScript^TM^ RT kit (Takara Biotech, Dalian, China). For expression analysis of maize regulatory transgenes, wheat orthologous regulatory and structural genes involved in the anthocyanin biosynthesis pathway were quantified by qRT-PCR in an ABI 7500 Real-Time PCR System (Life Technologies, Burlington, Ontario, Canada). A 20-μL reaction volume (SYBR PrimeScript RT-PCR Kit, Takara Biotech, Dalian, China) containing 10 μL 2 × SYBR Premix Ex Taq, 2 μL first-strand cDNA, 0.3 μL primer mix (10 μM), 0.4 μL ROX Reference DyeII, and 7.3 μL ddH_2_O were used. Primers targeting a product size of about 200 bp for the transgenes and wheat-related endogenous genes tested in this study for expression analysis are listed in Appendix A. Designed primer sequences for wheat *TaPpm*1 and *TaPpb1* and endogenous genes were selected from a previous study [34]. qRT-PCR was carried out by denaturing at 95 °C for 2 min, which was followed by 40 cycles at 95 °C for 5 s, 60 °C for 30 s, and 72 °C for 30 s. qRT-PCR reactions were normalized with *Triticum aestivum* adenosine diphosphate ribosylation factor (*TaADP*) in wheat [57]. All biological samples were measured in triplicate. The relative mRNA level was calculated using the 2-∆∆CT method [58]. The transgenic wheat line AL-45 with both transgenes was used as a control to calculate the fold-change expression of *ZmC1* and *ZmR* genes among transgenic lines. Fielder was used as a control to calculate the fold-change expression of the two orthologous and six structural wheat genes in the anthocyanin biosynthesis pathway among the transgenic lines as well as the wild types. The differences among the transgenic wheat lines were separated using Tukey’s HSD test at a 0.05 probability level in Statistix 8.1 [59]. Error bars above mean values reveal the Tukey’s values for the transgenic lines.

### 4.7. Quantitative Analysis of Anthocyanin Content

Fresh leaves were harvested using liquid nitrogen from the three types of transgenic wheat lines and the wild-types Fielder and Jingdong18. Similarly, about 25 dried seeds of each transgenic line with the wild types were also used for anthocyanin extraction. The fresh-frozen leaves and dried seeds were crushed using a mixer mill (MM 400, Retsch Technology, Haan, Germany) with a zirconia bead for 1.5 min at 30 Hz, and 100 mg powder was weighed and extracted overnight at 4 °C with 70% aqueous methanol. The supernatant was generated by centrifuging the re-vortexed mixture at 10,000× *g* for 10 min followed by filtering (0.22 μm pore size, ANPEL, Shanghai, China, www.anpel.com.cn/) before LC-MS analysis.

The sample extracts were analyzed using an LC-ESI-MS/MS system (HPLC, Shim-pack UFLC SHIMADZU CBM20A system, www.shimadzu.com.cn/, MS, Applied Biosystems 4000 Q TRAP, www.appliedbiosystems.com.cn/) as previously described [60]. The analytical conditions were as follows: HPLC, column, shim-pack VP-ODS C18 (pore size 5.0 μm, length 2 × 150 mm), solvent system, water (0.04% acetic acid): acetonitrile (0.04% acetic acid), gradient program, 95:5 V/V at 0 min, 5:95 V/V for 20.0 min, 5:95 V/V at 22.0 min, 95:5 V/V at 22.1 min, and 95:5 V/V at 28.0 min, flow rate, 0.25 mL min^‒1^, temperature, 40 °C, injection volume: 5 μL. The commercial standard cyanidin 3-*O*-glucoside used in this test was purchased from the New Asiatic Pharmaceuticals Co., Ltd., Shanghai, China. Gradient grades of methanol, acetonitrile, and acetic acid were purchased from Merck Company, Darmstadt, Germany (www.merckchemicals.com). Each sample was tested for the content of cyanidin 3-O-glucoside in three replicates. Data was analyzed and differences among the transgenic wheat lines were separated using Tukey’s HSD test at a 0.05 probability level in Statistix 8.1 software [59]. Error bars above mean values reveal the standard deviation (SD) for the transgenic lines.

## 5. Conclusions

Maize-derived genes *ZmC1* and *ZmR* encoding MYB-type *ZmC1* and bHLH *ZmR* transcription factors were transferred into wheat individually and in combination. Three types of stable transgenic wheat lines were obtained. Fluorescence in situ hybridization analysis revealed that the transgenes integrated in the heterochromatin region near the pericentric chromosome in the stable transgenic wheat lines. The phenotypes from the combined and independent transgenes revealed differential levels of anthocyanin biosynthesis from vegetative to maturity stages. The combined overexpression of the two genes led to higher levels of anthocyanin accumulation in various wheat tissues. However, *ZmC1* was responsible for pigmentation in vegetative tissues at different developmental stages while *ZmR* endowed the purple color at embryonic development in the grains. High transcript levels of the two transgenes and the six wheat structural genes were detected in the transgenic wheat lines containing *ZmC1* and *ZmR*. The tested genes displayed differential transcript levels in the other two transgenic wheat lines containing *ZmC1* or *ZmR*.

Our findings suggested that the regulatory TFs were direct activators regulating the expression of the structural or pathway genes and resulted in increased anthocyanin in a tissue-specific manner in wheat. Moreover, differential transcript levels in the light exposed and control seeds in the same spike of *ZmR* transgenic lines suggested that light exposure increases anthocyanin biosynthesis. Quantitative analysis of anthocyanin content in the three types of transgenic wheat plants suggests that the combined overexpression of *ZmC1* and *ZmR* led to high levels of anthocyanin production. In addition, knowledge of tissue-specific anthocyanin biosynthesis in wheat suggests that the two TFs can be used as phenotypic markers in wheat genetic transformations during the selection of other linked transgenes.

## Figures and Tables

**Figure 1 ijms-20-05806-f001:**
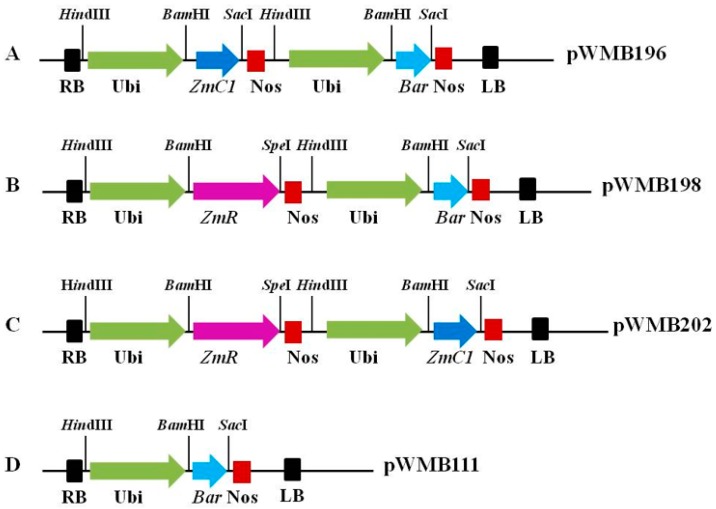
T-DNA regions of the expression vectors carrying maize *ZmC1* and *ZmR* genes for wheat transformation. (**A**) Vector pWMB196 contains the expression cassettes of *ZmC1* and *Bar* genes. (**B**) Vector pWMB198 contains the expression cassettes of *ZmR* and *Bar* genes. (**C**) Vector pWMB202 contains the expression cassettes of *ZmC1*, and *ZmR* genes. (**D**) Vector pWMB111 contains the expression cassette of *Bar* gene. Ubi: mazie ubiquitin promoter. Nos: *Agrobacterium* nopaline synthase terminator. LB: left border. RB: right border. *Bar*: bialaphos resistance gene.

**Figure 2 ijms-20-05806-f002:**
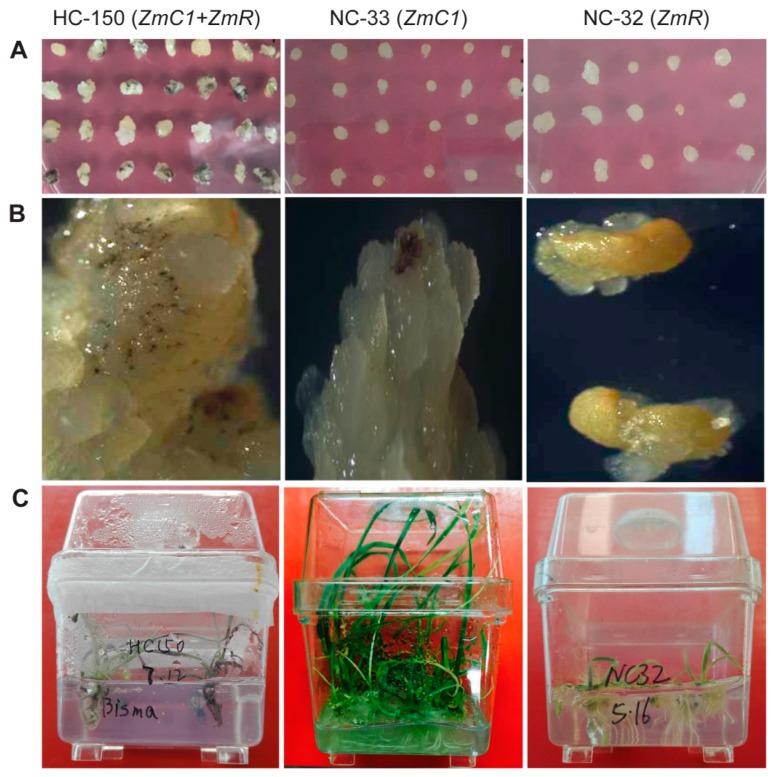
Anthocyanin phenotype in the transformed immature embryos (**A**), embryonic calluses (**B**), and regenerated plantlets (**C**) of wheat using different expression vectors. Experiment HC-150 was set up for the co-transformation of expression vector pWMB202 (carrying *ZmC1* and *ZmR*) and pWMB111 (carrying *Bar* only). NC-33 and NC-32 were set up for the transformation of expression vectors pWMB196 (carrying *ZmC1* only) and pWMB198 (carrying *ZmR* only), respectively. Plantlet age: about 60 days after regeneration culture accompanying phosphinothricin (PPT) selection.

**Figure 3 ijms-20-05806-f003:**
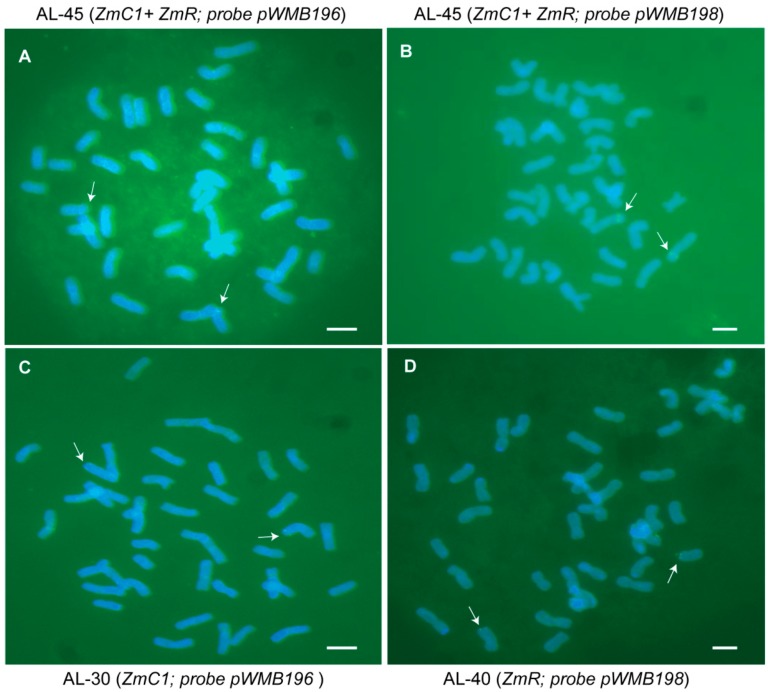
Integration detection of transgenes in the stable transgenic wheat lines by fluorescence in situ hybridization analysis (FISH) using pWMB196 and pWMB198 as probes. (**A**) The two hybridization signals indicated the integration of *ZmC1* in the line AL-45 was transformed with vector pWMB202. (**B**) The two hybridization signals indicated the integration of *ZmR* in the line AL-45 was transformed with vector pWMB202. (**C**) The two hybridization signals indicated the integration of *ZmC1* in the lines AL-30 was transformed with vector pWMB196. (**D**) The two hybridization signals indicated the integration of *ZmR* in the line AL-40 was transformed with vector pWMB198. Scale bar = 10 μm.

**Figure 4 ijms-20-05806-f004:**
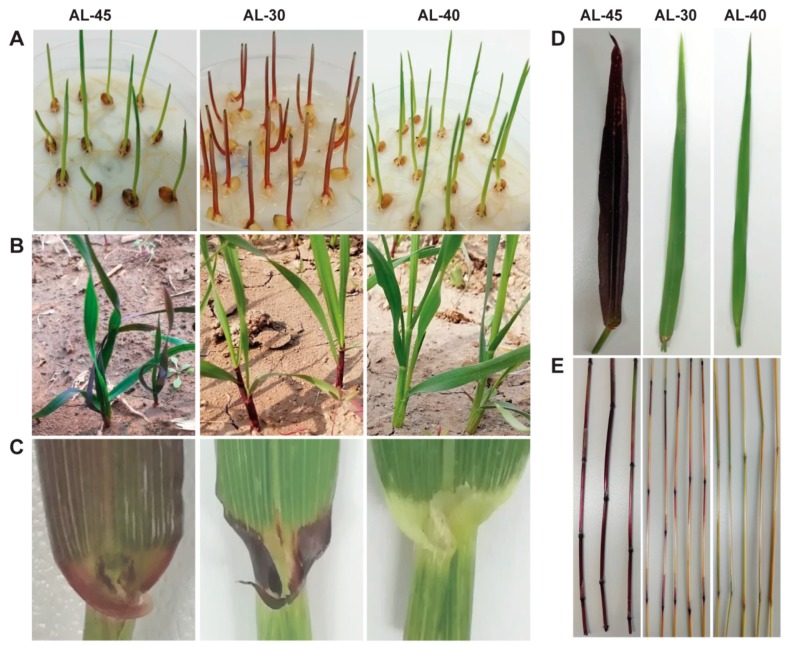
Anthocyanin phenotype in the various vegetative tissues of the three types of stable transgenic wheat lines. Transgenic line AL-45 containing *ZmC1* and *ZmR* was derived from the co-transformation with vectors pWMB202 and pWMB111. Transgenic lines AL-30 containing *ZmC1* only and AL-40 containing *ZmR* only were derived from the transformation with vectors pWMB196 and pWMB198, respectively. (**A**) coleoptiles, (**B**) seedlings, (**C**) auricles, (**D**) leaves, and (**E**) stems.

**Figure 5 ijms-20-05806-f005:**
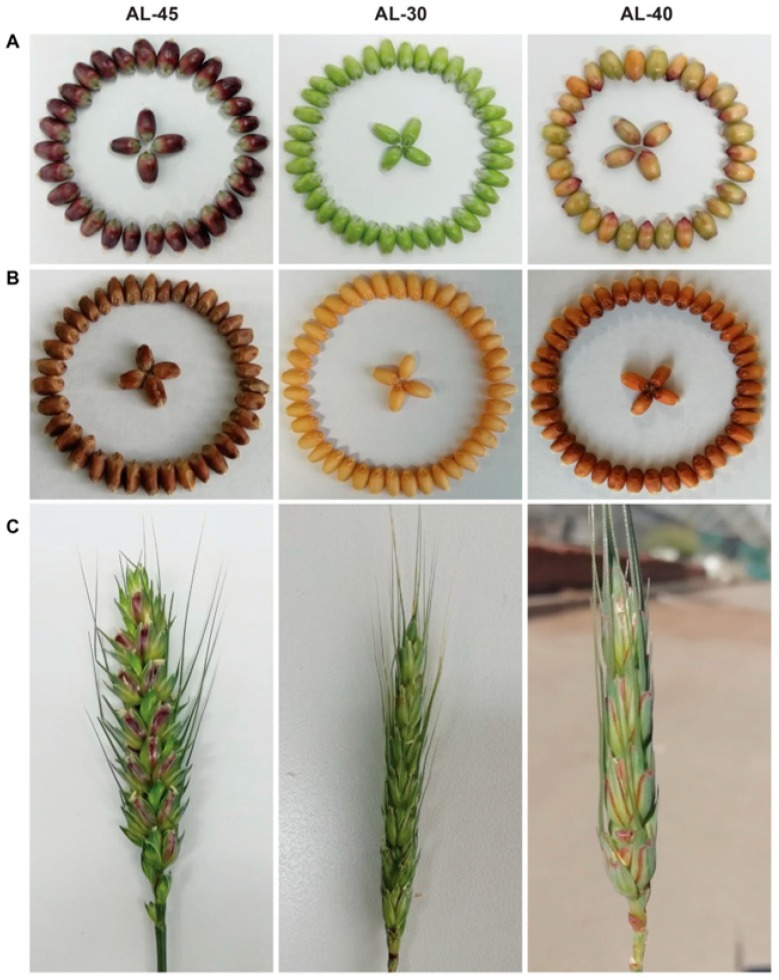
Anthocyanin accumulation in the reproductive tissues of the three types of stable transgenic wheat lines. Transgenic line AL-45 containing *ZmC1* and *ZmR* was derived from the co-transformation with vectors pWMB202 and pWMB111. Transgenic lines AL-30 containing *ZmC1* only and AL-40 containing *ZmR* only were derived from the transformation with vectors pWMB196 and pWMB198, respectively. (**A**) Immature seeds. (**B**) Dried seeds. (**C**) Spikes.

**Figure 6 ijms-20-05806-f006:**
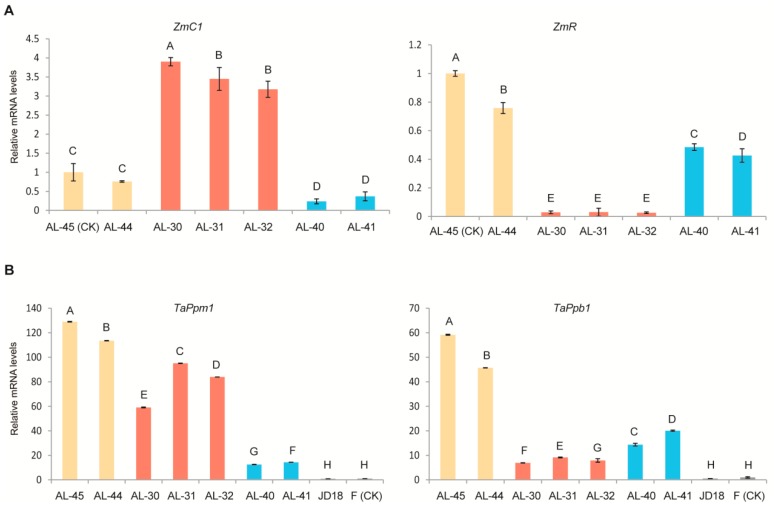
Expression analysis of maize *ZmC1* and *ZmR* genes (**A**), and their corresponding wheat homologous genes *TaPpm1* and *TaPpb1* (**B**) in the three types of stable transgenic wheat lines by quantitative real-time PCR (qRT-PCR). Transcript levels in the leaves of the three types of transgenic lines of AL-45 and AL-44 containing *ZmC1* and *ZmR* (yellow columns), AL-30, AL-31, and AL-32 containing *ZmC1* only *(*red columns), and AL-40 and AL-41 containing *ZmR* only (blue columns). Wild type plants of Fielder (F) and Jingdong18 (JD18) (black columns) were determined in three biological replicates. CK stands for the material used as a reference to calculate the relative expression level. Vertical bars are mean ± SD. Different letters in columns indicate statistically significant differences (*p* < 0.05).

**Figure 7 ijms-20-05806-f007:**
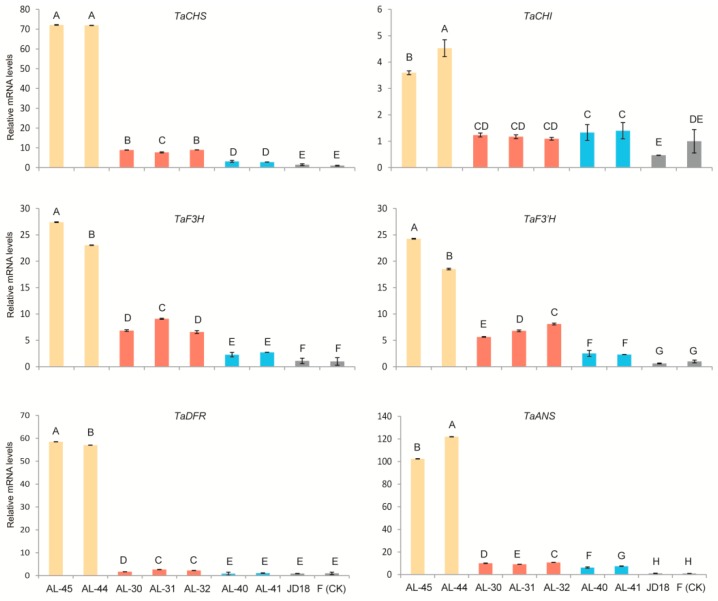
Expression analysis of six wheat native structural genes (*TaCHS*, *TaCHI*, *TaF3H*, *TaF3′H*, *TaDFR*, and *TaAN*S) involved in the anthocyanin biosynthesis pathway in the leaves of the three types of stable transgenic wheat lines by quantitative real-time PCR (qRT-PCR). Transcript levels in the leaves of the three types of transgenic lines AL-45 and AL-44 containing *ZmC1* and *ZmR* (yellow columns), AL-30, AL-31, and AL-32 containing *ZmC1* only (red columns), AL-40 and AL-41 containing *ZmR* only (blue columns), as well as the two wild types Fielder (F) and Jingdong18 (JD18) (black columns) were determined in three biological replicates. CK stands for the material used as a reference to calculate the relative expression level. Vertical bars are mean ± SD. Different letters in columns indicate statistically significant differences (*p* < 0.05).

**Figure 8 ijms-20-05806-f008:**
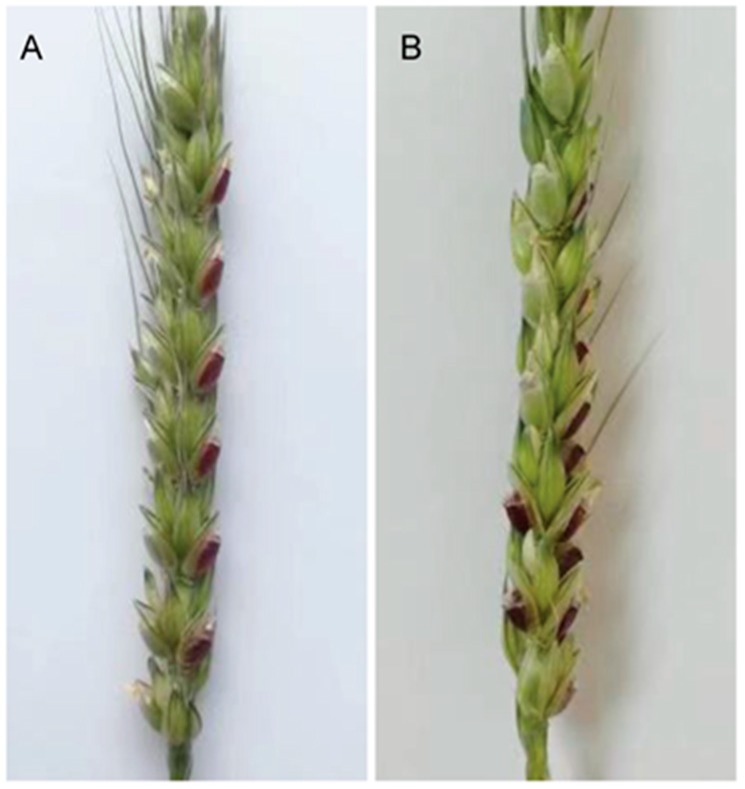
On the basis of phenotypic characterization, high anthocyanin accumulation was found in the developing seeds of *ZmR* containing stable transgenic lines, which were exposed to strong light by removing their lemma. (**A**) A spike from transgenic line AL-40. The seeds in one side exposed to strong light showed a purple color and the seeds in the other side closely protected with lemma showed a green color. (**B**) A spike from transgenic line AL-41. The seeds exposed to strong light appeared purple and the control seeds appeared green.

**Figure 9 ijms-20-05806-f009:**
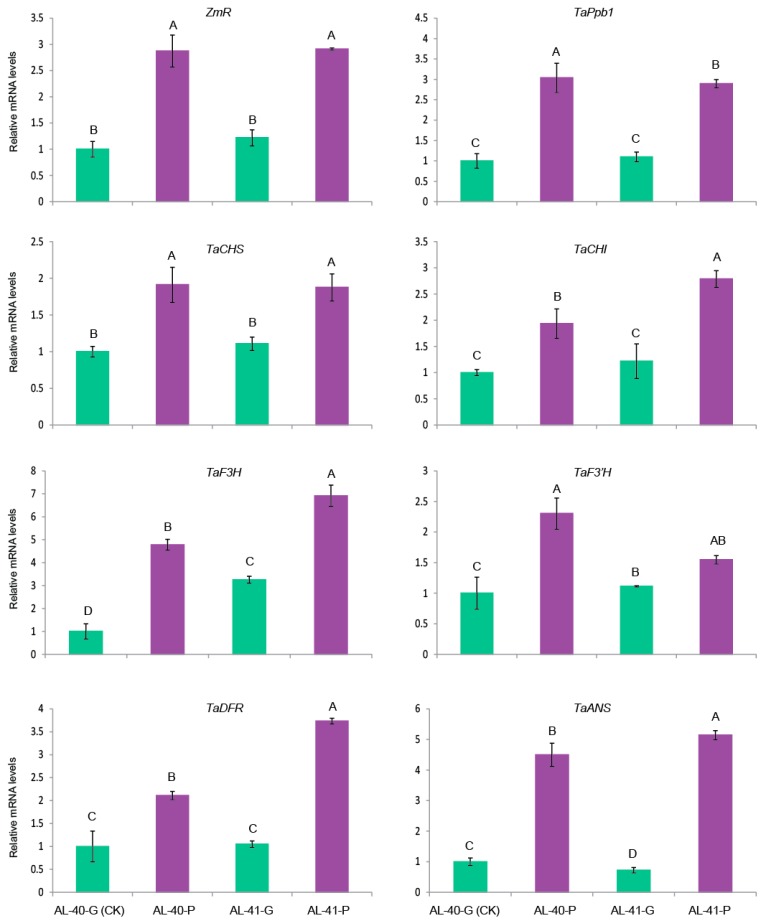
Expression profiles of maize *ZmR*, its wheat homologue *TaPpb1*, and six structural genes (*TaCHS*, *TaCHI*, *TaF3H*, *TaF3′H*, *TaDFR*, and *TaANS*) involved in anthocyanin biosynthesis in the light exposed seeds (AL-40-P and AL-41-P in purple color columns) and control seeds (AL-40-G and AL-41-G in green color columns) in the same spike in the two *ZmR* containing stable transgenic lines AL-40 and AL-41. CK stands for the sample used as a reference to calculate the relative expression level. Vertical bars are mean ± SD. Different letters in columns indicate statistically significant differences (*p* < 0.05).

**Figure 10 ijms-20-05806-f010:**
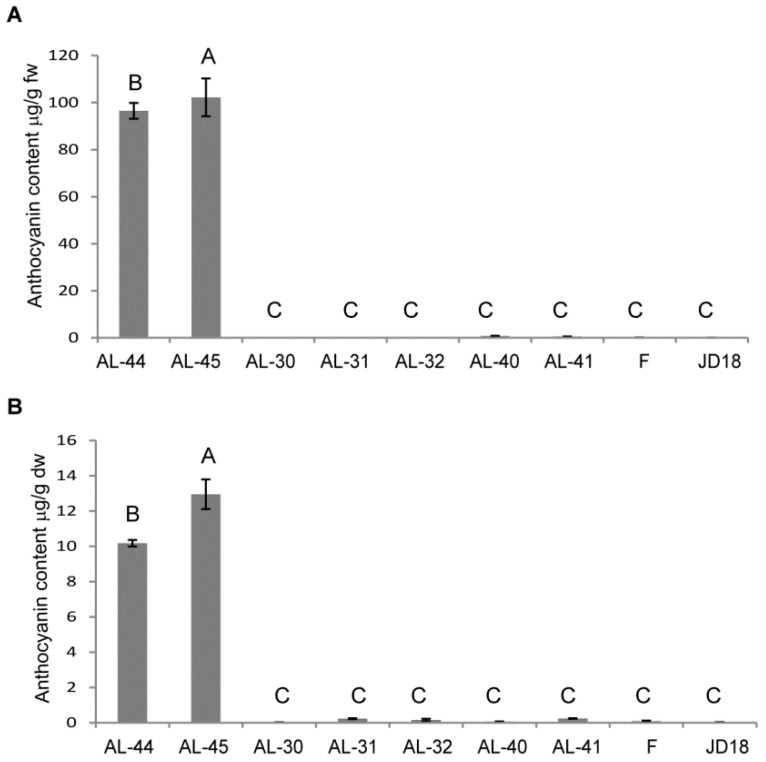
Quantitative analysis of total anthocyanin content in the three types of transgenic lines:AL-45 and AL-44 containing *ZmC1* and *ZmR*, AL-30, AL-31, and AL-32 containing *ZmC1* only, AL-40 and AL-41 containing *ZmR* only, and wild-type plants Fielder (F) and Jingdong18 (JD18). (**A**) Total anthocyanin contents in leaves, in microgram per gram of fresh weight. (**B**) Total anthocyanin contents in kernels, in microgram per gram of dry weight. Vertical bars are mean ± SD. Different letters in columns indicate statistically significant differences (*p* < 0.05).

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
