# Peer review of "Overexpression of Maize ZmC1 and ZmR Transcription Factors in Wheat Regulates Anthocyanin Biosynthesis in a Tissue-Specific Manner"

_ijms, 2019, doi:10.3390/ijms20225806_

Round 1

Reviewer 1 Report

The manuscript “Expression of Maize ZmC1 and ZmR Transcription Factors in Wheat Regulates Anthocyanin Biosynthesis in Tissue-specific Manner” is a revised version of the previously submitted manuscript “Overexpression of Maize ZmC1 and ZmR Transcriptional Factors in Wheat Regulates Anthocyanin Biosynthesis in Different Tissues”. The authors have dramatically revised the text, performed additional experiments, included necessary controls.  The authors have addressed all previous comments, except the altering the units used for the light intensity. I insist on rechecking this information.

Thorough revision of the text is still required to improve the language.

Symbols for genes have to be italicized (it is corrected by authors), but symbols for proteins should not be italicized.

Author Response

Dear Reviewers,

Thanks a lot for your thoughtful comments on our manuscript entitled “Overexpression of maize ZmC1 and ZmR transcription factors in wheat regulates anthocyanin biosynthesis in tissue-specific manner” (ijms-603105). We have carefully revised the manuscript according to the comments generated by reviewers and journal requirements. All the changes are highlighted with red/blue color in the modified manuscript. Our point-by-point responses to the comments of reviewers and editors are presented below.

1) Symbols for genes have to be italicized (it is corrected by authors), but symbols for proteins should not be italicized.

Response: Thank you a lot for your comment and suggestion. We have been corrected the writings for the symbols of genes, proteins, and plasmids in the revised manuscript.

Thanks again for your time in reviewing our manuscript.

Sincerely yours,

Bisma Riaz and Xingguo Ye

bisma.riaz75@gmail.com, yexingguo@caas.cn

Institute of Crop Sciences

Chinese Academy of Agricultural Sciences

Reviewer 2 Report

The work was simple, elegant and interesting. The authors used a range of different methods to analyze the activity of regulatory genes of anthocyanins biosynthesis - MYB-type ZmC1 and bHLH-type ZmR of maize. These genes were transferred via agrobacterial transformation into the bread wheat. As a result, the authors confirmed some existing hypotheses regarding the specific features of the accumulation of anthocyanin pigments in various plant tissues due to the tissue-specific activity of the MYB and bHLH genes.

However, extreme neglect in the description of the literature data and in the description of the experimental part is noted in the work. The paper is overall interesting and deserves publication, but there are some major flaws that should be taken into consideration before final acceptance. 

63-65 «purple plant 1 (PL1) … colored aleurone 1 (C1) … R1 (red) … B1 (booster) … PAC1 (pale aleurone color 1)» - it is necessary to bring the signatures to uniformity: either the name of the gene first, the decoding in brackets, or vice versa.

75-76 «Several wheat genes which control anthocyanin accumulation in its various parts have been documented» - not enough literature references.

76-77 «Only the red grain (R) gene encoding MYB TF promoted the accumulation of the anthocyanins in wheat seeds [30,31]» - this is not true. First, the MYB gene in the original article [30] is called differently (Tamyb10). Secondly, in 2019 (Strygina, Ksenia V., and Elena K. Khlestkina. "Structural and functional divergence of the Mpc1 genes in wheat and barley." BMC evolutionary biology 19.1 (2019): 45.) for the MYB wheat genes, the name was put in order according to the primary source (Li, W. L., et al. "Genomic mapping of defense response genes in wheat." Theoretical and Applied Genetics 98.2 (1999): 226-233.). Thus, “Mpc1” instead if “C1” or “Ant1” was used in the gene names according to the first description and designation (Mpc1) of the C1 homologs in Triticeae tribe. Third, article [31] is devoted to the biosynthesis of proanthocyanidins in a barley grain. Fourth, in 2014 (Shoeva, Olesya, Elena Gordeeva, and Elena Khlestkina. "The regulation of anthocyanin synthesis in the wheat pericarp." Molecules 19.12 (2014): 20266-20279.), the regulator of the synthesis of anthocyanins in wheat pericarp is TaMyc-1 (or TaMyc-A1 in accordance with Strygina, Ksenia V., and Elena K. Khlestkina. "Myc-like transcriptional factors in wheat: structural and functional organization of the subfamily I members." BMC plant biology 19.1 (2019): 50.).

77-79 «Other anthocyanin biosynthesis associated genes including purple anther (Pan), purple glume (Pg), purple leaf blade (Plb), purple pericarp (Pp), and red coleoptile (Rc) have also been documented [32]» - the description of known information about transcription factors coding these genes is missed.

79-81 «Even though C1 and R genes are also presented in wheat regulating the pigmentation, in most of wheat genotypes they just have a low expression (data unpublished) because of weak promoters» - wrong. For MYB and bHLH genes that control the synthesis of anthocyanins in wheat, there is an article (Jiang, Wenhui, et al. "Two transcription factors TaPpm1 and TaPpb1 co-regulate anthocyanin biosynthesis in purple pericarps of wheat." Journal of experimental botany 69.10 (2018): 2555-2567.), which describes the structural differences of the promoter.

105 Figure 1 – there is no description of abbreviations

109-111 – lack of phenotypic description. Why were these genotypes chosen? What mutations in the regulatory genes MYB and bHLH that control anthocyanins synthesis in wheat pericarp are noted for these genotypes? The final result of work directly depends on this.

130-131 «for two and half months» - how many days?

168-170 «MYB-type ZmC1 expressed only in vegetative tissues while bHLH-type ZmR expressed only in the reproductive tissues (Figure 4 and Figure 5)» - there are no graphs with gene expression in these figures.

174, 358 «MWB» - MBW

191 «TaC1 and TaR» - firstly, incorrect gene names. Secondly, to which of the homeologous copies of the genes were the primers selected? It is necessary to additionally align. The explanation of the results directly depends on this.

195-198 «Results indicated that ZmC1 and ZmR expressed in a dramatic high level in the three types of transgenic lines, especially the ZmC1 in the transgenic lines AL-30, AL-41, and AL-42 followed by AL-45,and AL-44, and the ZmR in the transgenic lines AL-45 and AL-44 followed by AL-31 and AL-32 (Figure 6)» - the level of expression cannot be dramatically high, because the authors have never made a statistical analysis of the results (by the way, why?)

215 «CHS, CHI, F3H, F3'H, DFR, and ANS (Supplementary Figure S5)» - in Supplementary Figure S5 does not describe which expression level of which copies of these genes was measured. There are no references to the NCBI, from which it is not clear which alleles the primers were selected for. In addition, the style of describing gene names is changing - for example, CHS and TaCHS.

223-226 «Nevertheless, no big difference on the expression levels ...» - there is no statistical analysis of the results.

248 «Figure 8. High anthocyanin accumulation…» - no measurements of the relative level of anthocyanins in the grain were made. The phrase is incorrect.

266 Figure 9 – there is no statistical analysis of the results.

270 2.7. Anthocyanin contents determination in the three types of transgenic wheat plants – there is no data on the content of specific anthocyanins in specific tissues. It is not clear how many replicates were used. No statistical processing.

291 - The discussion should be adjusted in accordance with the data on the allelic state of the TaMyc and TaMpc genes, a complete description of the results and its statistical analysis.

475 – Why did you use ethidium bromide in your work?

499 – Why did you choose only one reference gene for these tissues, and this particular one? Also needed are tables with data illustrating the level of gene expression.

501 - Tables with data illustrating the level of anthocyanins are needed.

541 - TaADP is not included in the list of abbreviations

Figure S2 «C: Roots» - C: roots

Table S1. – there is no description of the length of the product. It is unclear whether the authors developed all the primers by themselves or used someone else's. It is not clear whether primers were selected for all paralogous and homeologous copies of wheat genes, or to the particular one. It is not clear which alleles primers were selected for - GenBank links are needed.

Author Response

Dear Reviewer,

Thanks a lot for your thoughtful comments on our manuscript entitled “Overexpression of maize ZmC1 and ZmR transcription factors in wheat regulates anthocyanin biosynthesis in tissue-specific manner” (ijms-603105). We have carefully revised the manuscript according to the comments. All the changes are highlighted with red/blue color in the modified manuscript. Our point-by-point responses to the comments are presented below.

63-65 «purple plant 1 (PL1) … colored aleurone 1 (C1) … R1 (red) … B1 (booster) … PAC1 (pale aleurone color 1)» - it is necessary to bring the signatures to uniformity: either the name of the gene first, the decoding in brackets, or vice versa.

Response: Thank you so much for the nice suggestion. We have corrected them accordingly in the revised manuscript.

75-76 «Several wheat genes which control anthocyanin accumulation in its various parts have been documented» - not enough literature references.

Response: Thank you for your comment. We have revised the introduction part of the manuscript by adding more information and references.

76-77 «Only the red grain (R) gene encoding MYB TF promoted the accumulation of the anthocyanins in wheat seeds [30,31]» - this is not true. First, the MYB gene in the original article [30] is called differently (Tamyb10). Secondly, in 2019 (Strygina, Ksenia V., and Elena K. Khlestkina. "Structural and functional divergence of the Mpc1 genes in wheat and barley." BMC evolutionary biology 19.1 (2019): 45.) for the MYB wheat genes, the name was put in order according to the primary source (Li, W. L., et al. "Genomic mapping of defense response genes in wheat." Theoretical and Applied Genetics 98.2 (1999): 226-233.). Thus, “Mpc1” instead if “C1” or “Ant1” was used in the gene names according to the first description and designation (Mpc1) of the C1 homologs in Triticeae tribe. Third, article [31] is devoted to the biosynthesis of proanthocyanidins in a barley grain. Fourth, in 2014 (Shoeva, Olesya, Elena Gordeeva, and Elena Khlestkina. "The regulation of anthocyanin synthesis in the wheat pericarp." Molecules 19.12 (2014): 20266-20279.), the regulator of the synthesis of anthocyanins in wheat pericarp is TaMyc-1 (or TaMyc-A1 in accordance with Strygina, Ksenia V., and Elena K. Khlestkina. "Myc-like transcriptional factors in wheat: structural and functional organization of the subfamily I members." BMC plant biology 19.1 (2019): 50.).

Response: Thank you so much for your comment and nice suggestions. We have well modified the text in the revised manuscript.

77-79 «Other anthocyanin biosynthesis associated genes including purple anther (Pan), purple glume (Pg), purple leaf blade (Plb), purple pericarp (Pp), and red coleoptile (Rc) have also been documented [32]» - the description of known information about transcription factors coding these genes is missed.

Response: Thanks a lot for the valuable comment. We have included this information in the revised manuscript.

79-81 «Even though C1 and R genes are also presented in wheat regulating the pigmentation, in most of wheat genotypes they just have a low expression (data unpublished) because of weak promoters» - wrong. For MYB and bHLH genes that control the synthesis of anthocyanins in wheat, there is an article (Jiang, Wenhui, et al. "Two transcription factors TaPpm1 and TaPpb1 co-regulate anthocyanin biosynthesis in purple pericarps of wheat." Journal of experimental botany 69.10 (2018): 2555-2567.), which describes the structural differences of the promoter.

Response: Thank you for your valuable suggestion. We have modified the text in the introduction part of our manuscript.

105 Figure 1 – there is no description of abbreviations.

Response: Thank you for your suggestion. We have revised the legend of Figure 1.

109-111 – lack of phenotypic description. Why were these genotypes chosen? What mutations in the regulatory genes MYB and bHLH that control anthocyanins synthesis in wheat pericarp are noted for these genotypes? The final result of work directly depends on this.

Response: We are appreciated to your comments. We have included the phenotypic description of these genotypes in the revised manuscript. Previous studies showed that mutations are often detected in the promoter and coding regions of the regulatory genes MYB and bHLH of non-purple plant varieties.

130-131 «for two and half months» - how many days?

Response: Thanks. These were about 76 days. We have mentioned it in the revised manuscript.

168-170 «MYB-type ZmC1 expressed only in vegetative tissues while bHLH-type ZmR expressed only in the reproductive tissues (Figure 4 and Figure 5)» - there are no graphs with gene expression in these figures.

Response: Thank you for your kind suggestion. We have revised the text in the revised version of our manuscript.

174, 358 «MWB» - MBW.

Response: Sorry for the mistake. We have corrected it in the revised manuscript.

191 «TaC1 and TaR» - firstly, incorrect gene names. Secondly, to which of the homeologous copies of the genes were the primers selected? It is necessary to additionally align. The explanation of the results directly depends on this.

Response: Thank you for your nice suggestion. We have modified this part by the comment in the revised manuscript.

195-198 «Results indicated that ZmC1 and ZmR expressed in a dramatic high level in the three types of transgenic lines, especially the ZmC1 in the transgenic lines AL-30, AL-41, and AL-42 followed by AL-45,and AL-44, and the ZmR in the transgenic lines AL-45 and AL-44 followed by AL-31 and AL-32 (Figure 6)» - the level of expression cannot be dramatically high, because the authors have never made a statistical analysis of the results (by the way, why?)

Response: Thanks for the constructive comment. We have performed Tukey’s HSD test and included it in the revised manuscript.

215 «CHS, CHI, F3H, F3'H, DFR, and ANS (Supplementary Figure S5)» - in Supplementary Figure S5 does not describe which expression level of which copies of these genes was measured. There are no references to the NCBI, from which it is not clear which alleles the primers were selected for. In addition, the style of describing gene names is changing - for example, CHS and TaCHS.

Response: Thank you for your nice suggestion and pointing out the mistake. We have included this detail in the revised manuscript.

223-226 «Nevertheless, no big difference on the expression levels ...» - there is no statistical analysis of the results.

Response: We are grateful to your nice suggestion. We have performed the statistical analysis in the revised manuscript.

248 «Figure 8. High anthocyanin accumulation…» - no measurements of the relative level of anthocyanins in the grain were made. The phrase is incorrect.

Response: Thank you for your suggestion. We have corrected it in the revised manuscript.

266 Figure 9 – there is no statistical analysis of the results.

Response: Thank you for your suggestion. We have performed the Tukey’ test in the revised manuscript.

270 2.7. Anthocyanin contents determination in the three types of transgenic wheat plants – there is no data on the content of specific anthocyanins in specific tissues. It is not clear how many replicates were used. No statistical processing.

Response: Thank you for the adequate suggestion. The major determined anthocyanin content was cyanidin 3-O-glucoside and presented as total anthocyanin contents. We have performed Tukey test analysis on the anthocyanin data and detail has been included in the revised manuscript.

291 - The discussion should be adjusted in accordance with the data on the allelic state of the TaMyc and TaMpc genes, a complete description of the results and its statistical analysis.

Response: Thank you for your nice suggestions. We have carefully modified it in the revised manuscript.

475 – Why did you use ethidium bromide in your work?

Response: Thank you for pointing out the mistake. We have corrected the text in the revised manuscript.

499 – Why did you choose only one reference gene for these tissues, and this particular one? Also needed are tables with data illustrating the level of gene expression.

Response: We choose only one reference gene TaADP, which is widely employed in wheat; as samples were collected at only one development stage and tissues. It was not necessary to test the expression level of the reference gene; therefore, we didn’t show the expression level of TaADP gene in wheat tissues.

501 - Tables with data illustrating the level of anthocyanins are needed.

Response: Thanks a lot for the suggestion. An new Table S1 containing the data illustrating the level of anthocyanins has been added in the Supplementary Materials.

541 - TaADP is not included in the list of abbreviations.

Response: Thanks for the comment. We have added the explanation of TaADP (Triticum aestivum adenosine diphosphate ribosylation factor) in the list of abbreviations.

Figure S2 «C: Roots» - C: roots

Response: Thanks for pointing out the mistake. We have corrected it in the revised manuscript.

Table S1. – there is no description of the length of the product. It is unclear whether the authors developed all the primers by themselves or used someone else's. It is not clear whether primers were selected for all paralogous and homeologous copies of wheat genes, or to the particular one. It is not clear which alleles primers were selected for - GenBank links are needed.

Response: Thank you for your comment. We have included this detail in the revised manuscript.

Thanks again for your time in reviewing our manuscript.

Sincerely yours,

Bisma Riaz and Xingguo Ye

bisma.riaz75@gmail.com, yexingguo@caas.cn

Institute of Crop Sciences

Chinese Academy of Agricultural Sciences

Reviewer 3 Report

The manuscript by Riaz et al. describes phenotypic and molecular characterization of wheat transgenic lines overexpressing maize transcription factors (TFs), R2R3 MYB C1 and bHLH factor R, either individually or in combination.

major comments:

I am looking at this manuscript for the first time and I have several comments. This version of the manuscript is not ready for publication. The authors need to address these issues in the revised version.

1.Why did the authors use two different cultivars for transformation ?

2. I do not think FISH is that important for this study to show the transgene integration. Some of the pictures are not clear. “strong green hybridization” signal is not clear in all pictures. PCR/qPCR analysis and phenotypic characterization (pigment accumulation transgenic plants) confirm the integration of transgenes.

3. line 167-169: Both ZmC1 and ZmR appear to be driven by a constitutive promoter, supposed to express in all tissues. What do these statements mean here? The observed phenotype could be due the presence or absence of other interacting partners in that tissue.

4. line 171 AL45: Do not see any pigments in coleoptile ?

5. I do not understand the rationale behind measuring the expression of TaC1 and TaR ? Please explain?

6. How did the authors measure relative expression of ZmC1 and ZmR in transgenic lines ? This is a heterologous system (the wild-type plants/ cultivars used in this study do not have that gene). Could you provide an explanation or edit the text.

7. Surprised to see DFR and ANS are not induced in the C1 and R transgenic lines. Are the tissue samples chosen for this experiments ok ?

8. Both ZmC1 and ZmR appear to be driven by a constitutive promoter. How did the expression of ZmR change in response to light?

9. Discussion: Many statements made in this section are confusing, and not correct. I would like authors to provide an explanation.

a. line 311-315 In the present study, transgenic lines ……….pigmentation in vegetative tissues like coleoptiles.”

b. line 337-338 “Unlike the other species, we conclude….. AL-30 and AL-40 (Figure 4, Figure 5). How did the authors come to this conclusion??

c. line 346-347 “This is consistent with our result ……. highest transcripts level of ZmR (Figure 5).” How these results are consistent with that of Medicago TT8 regulation, could the authors explain?

d. line 349-351 “Whereas, the expression level of wheat…… line AL-45 as expected (Figure 5). How and why these results are relevant to this study?” Pl. explain?

e. “However, LBGs were not regulated …. the same trend as that in maize and Arabidopsis.” This could be due to the tissue used for expression analysis. For example, did the authors check the expression of LBG in coleoptile of AL30 or seeds of AL40 transgenic lines ?

10. Conclusion: line 394: transgenes NOT target genes. Why the endogenous TF genes are important here in conclusion? Is the observed phenotype due the induced expression of endogenous C1 or R?

minor comments:

Figure 1: Define some of these terminologies used in the figure: For example What is Ubi, ubiquitin promoter from maize or some other plant ?

line 49-50 dihydroflavonol 4-reductase is DFR NOT F3R.

line 109   many purple spots

line 188 Expression profiling of the two transgenes NOT target genes. also correct this in Abstract (line 20).

Error bars in figures: Define the error bars in Figures 6, 7 and 9 (SD or SE) ???

Figure 9 legend : light exposure seeds (AL-40-G and AL-41-G) and control seeds (AL-40-G and AL-41-G) in the same…..(Which one is control because both are “G” ??)

Figure 10: no statistics for this figure.

Supplement Fig. 2: why the background of some pictures is pink or purple? Difficult to see the phenotype.

Author Response

Dear Reviewer,

Thanks a lot for your thoughtful comments on our manuscript entitled “Overxpression of maize ZmC1 and ZmR transcription factors in wheat regulates anthocyanin biosynthesis in tissue-specific manner” (ijms-603105). We have carefully revised the manuscript according to the comments generated by reviewers and journal requirements. All the changes are highlighted with red/blue color in the modified manuscript. Our point-by-point responses to your comments are presented below.

Major comments:

Why did the authors use two different cultivars for transformation?

Response: In this study, we used two wheat cultivars to develop transgenic wheat germplasm with enhanced anthocyanin, in which Fielder is a spring and easy transformable genotype, and Jingdong18 is a winter and difficult transformable genotype. Because anthocyanin accumulation via expressing the two maize transcription factors seriously affects plant regeneration and growth of wheat, we only obtained five transgenic plants from Jingdong18 carrying the both transgenes ZmC1 and ZmR. Unfortunately, we didn’t harvest any transgenic plants from Fielder carrying the both transgenes except two plants with ZmR and sixteen plants carrying ZmC1 gene, respectively. In addition, wheat transformation is still a tough task; we normally use more cultivars in our transformation experiments.

I do not think FISH is that important for this study to show the transgene integration. Some of the pictures are not clear. “strong green hybridization” signal is not clear in all pictures. PCR/qPCR analysis and phenotypic characterization (pigment accumulation transgenic plants) confirm the integration of transgenes.

Response: Thank you very much for your comment. FISH analysis can visibly check the homozygous integration of the transgene and the result confirmed by FISH is also very solid even though the signals are some small or weak. Therefore, we often detected our transgenic wheat plants by FISH analysis for publications such as Liang et al., J Exp Bot, 2019, 70(5): 1539-1551; Liu et al., J Integra Agri, 2019, doi: 10.1016/S2095-3119(19)62601-8.

line 167-169: Both ZmC1 and ZmR appear to be driven by a constitutive promoter, supposed to express in all tissues. What do these statements mean here? The observed phenotype could be due the presence or absence of other interacting partners in that tissue.

Response: Thank you for your comment. It is the phenotypic characterization of the transgenic lines and it is possible that the observed phenotype could be due the presence or absence of other interacting partners in that tissue.

line 171 AL45: Do not see any pigments in coleoptile?

Response: Transgenic line AL-45 also showed light pigmentation on the coleoptiles.

I do not understand the rationale behind measuring the expression of TaC1 and TaR ? Please explain?

Response: Expression level of wheat TF TaPpm1 and TaPpb1 (also termed as TaC1 and TaR) were measured in the transgenic wheat lines to evaluate whether the overexpression of maize MYB and bHLH TF can also enhance the expression level of wheat corresponding TF because anthocyanin accumulation was observed and determined in the transgenic wheat lines.

How did the authors measure relative expression of ZmC1 and ZmR in transgenic lines ? This is a heterologous system (the wild-type plants/ cultivars used in this study do not have that gene). Could you provide an explanation or edit the text.

Response: Thank you for your suggestion. We have carefully edited the text.

Surprised to see DFR and ANS are not induced in the C1 and R transgenic lines. Are the tissue samples chosen for this experiments ok?

Response: Thank you for your comment. In Arabidopsis, DFR and ANS are referred as late biosynthetic genes and only transcriptionally activate through MBW complex. They were not regulated in the both types of transgenic lines AL-30, AL-31, and AL-32 containing ZmC1 and AL-40 and AL-41 containing ZmR due to the absence of MBW complex, suggesting that the activation of LBGs in wheat showed the same trend as that in maize and Arabidopsis [2, 48]. In addition, the regulation role of the two genes has tissue-specificity, for example, ZmR transgenic lines displayed the purple phenotype only in reproductive tissues like seeds. Therefore, the samples for qRT-PCR were collected from leaf tissues.

Both ZmC1 and ZmR appear to be driven by a constitutive promoter. How did the expression of ZmR change in response to light?

Response: Thank you for your comment. qRT-PCR analysis indicated that ZmR expression changes in response to light. This phenomenon might be due to the accumulative effect on ZmR expression of wheat corresponding or structural genes in seeds under light condition. Light might promote the expression of some seed-specific genes and then promote the expression of ZmR in seed and anthocyanin biosynthesis in this tissue. Exact reason for this phenomenon needs to be investigated further.

Discussion: Many statements made in this section are confusing, and not correct. I would like authors to provide an explanation. line 311-315 In the present study, transgenic lines ……….pigmentation in vegetative tissues like coleoptiles.”

Response: Thanks for the comment. We have edited the phrase in the revised manuscript.

line 337-338 “Unlike the other species, we conclude….. AL-30 and AL-40 (Figure 4, Figure 5). How did the authors come to this conclusion??

Response: Thanks. MYB ZmC1 in wheat control the pigmentation in vegetative tissues, however ZmC1 in maize and OsC1 in rice regulated the anthocyanin biosynthesis in reproductive tissues like seeds [2, 42, 43]. In maize, bHLH type different b alleles (orthologous to TaPpb1), B-peru and B-I regulated the specific anthocyanin production in seeds and vegetative tissues, respectively. In Aegilops tauschii bHLH AetMYC modulated the anthocyanin synthesis in the coleoptile (vegetative tissue) while in wheat TaPpb1 or ZmR regulate the anthocyanin synthesis in seeds (reproductive tissue). We have modified the description.

line 346-347 “This is consistent with our result ……. highest transcripts level of ZmR (Figure 5).” How these results are consistent with that of Medicago TT8 regulation, could the authors explain?

Response: Thanks lot for the comment. In Medicago truncatula the bHLH factor MtTT8 regulated its own expression as part of a positive feedback loop, through an MBW complex, that ultimately contributes to the regulation of anthocyanin synthesis. We also observed that the expression level of bHLH ZmR was higher in the transgenic lines containing the both transgenes (ZmR and ZmC1) than that in the transgenic lines with only ZmR transgene. This phenomenon might be due to the formation of MBW complex bHLH factor in which ZmR regulated its own expression as part of a positive feedback loop.

line 349-351 “Whereas, the expression level of wheat…… line AL-45 as expected (Figure 5). How and why these results are relevant to this study?” Pl. explain?

Response: Thank you very much for the comment. We have edited the sentences in the revised manuscript.

“However, LBGs were not regulated …. the same trend as that in maize and Arabidopsis.” This could be due to the tissue used for expression analysis. For example, did the authors check the expression of LBG in coleoptile of AL30 or seeds of AL40 transgenic lines?

Response: Thanks for the nice suggestion. Yes, this could be also due to the tissues specific. We have modified the text in the revised manuscript.

Conclusion: line 394: transgenes NOT target genes. Why the endogenous TF genes are important here in conclusion? Is the observed phenotype due the induced expression of endogenous C1 or R?

Response: We are appreciated to your suggestion. We mainly proposed that the overexpression of the transgenes induces the observed phenotype by transcriptional activation of wheat structural genes involved in anthocyanin biosynthesis pathway. We have modified it in the revised manuscript.

Minor comments:

Figure 1: Define some of these terminologies used in the figure: For example What is Ubi, ubiquitin promoter from maize or some other plant?

Response: Thanks for the suggestion. In figure 1, “Ubi” represents the maize ubiquitin promoter and “Nos” represents the Agrobacterium nopaline synthase terminator. We have revised the legend of Figure 1.

line 49-50 dihydroflavonol 4-reductase is DFR NOT F3R.

Response: Thank you for pointing out the mistake. We have corrected it in the revised manuscript.

line 109   many purple spots

Response: Thanks for the comment. We have revised the text in the revised draft of manuscript.

line 188 Expression profiling of the two transgenes NOT target genes. also correct this in Abstract (line 20).

Response: Thanks for the adequate suggestion. We have corrected it in the revised manuscript.

Error bars in figures: Define the error bars in Figures 6, 7 and 9 (SD or SE) ???

Response: Thank you. Error bars showed the standard deviation (SD).

Figure 9 legend : light exposure seeds (AL-40-G and AL-41-G) and control seeds (AL-40-G and AL-41-G) in the same…..(Which one is control because both are “G” ??)

Response: Thank you for pointing out the mistake. We have corrected it in the revised manuscript.

Figure 10: no statistics for this figure.

Response: Thanks for the suggestion. We have modified the picture and performed the statistical analysis in the revised manuscript.

Supplement Fig. 2: why the background of some pictures is pink or purple? Difficult to see the phenotype.

Response: These pictures were taken in the greenhouse during the growing period of transgenic plants in pots with white background, and this figure shows some pink color due to the light reflection.

Thanks again for your time in reviewing our manuscript.

Sincerely yours,

Bisma Riaz and Xingguo Ye

bisma.riaz75@gmail.com, yexingguo@caas.cn

Institute of Crop Sciences

Chinese Academy of Agricultural Sciences

Reviewer 4 Report

Manuscript entitled “Expression of Maize ZmC1 and ZmR Transcription Factors in Wheat Regulates Anthocyanin Biosynthesis in Tissue-specific Manner” by Riaz et al describes the characterization of several wheat transgenic lines where two maize transcription factors (ZmC1 and ZmR), previously described as involved in the regulation of anthocyanin biosynthesis, are constitutively expressed. The authors have been able to generate several stable transgenic lines where these transcription factors are overexpressed alone or co-expressed. The transgenic plants obtained have been characterized by several approaches, including visual observation and quantification of anthocyanin levels, as well as the expression profiles of a set of genes include the ZmR1 and ZmR homologues in wheat, as well as a set of genes that code for anthocyanin biosynthesis pathway enzymes. The authors found significant increases in anthocyanin content in the leaves and seeds of plants with simultaneous constitutive expression of both transcription factors, while in the lines that overexpress these factors individually only a localized anthocyanin tissue accumulation is observed in ZmR1-overexpressing lines These results are consistent with the levels of expression of the anthocyanin biosynthesis related genes that are observed in each of the transgenic lines generated. Additionally, the authors show that light exposure in ZmR1-overexpressing lines increased anthocyanin seed content, as well as the expression of the anthocyanin-related genes previously tested.
The proposed approaches and the strategy used for the generation and characterization of the wheat transgenic lines seem both correct for the most part. The results obtained are interesting, and for the most part well discussed but, in my opinion, some issues need to be revised throughout the manuscript, particularly how the results obtained are presented in some figures, as well as some language expressions used by the authors.

Some comments about your paper:
a) The authors have generated transgenic lines where ZmC1 and ZmR are under the control of ubiquitin1 promoter, which is a strong and constitutive promoter. Because of this, I think that “expression of the ZmR1 or ZmR gene" is not an accurate term to describe the transgenic approaches used in this work. I must suggest to the authors that they state more clearly what has been done in the transgenic lines generated, like "overexpression of the X gene" or "constitutive expression of gene X". Please consider to rephrase this term in all the manuscript. Also, expressions like “together expression of ” (Abstract, lines 23-24 and line 381)are not correct, please change with another terms like “simultaneous overexpression”

b) Regarding Introduction:
- Lines 36-38: Please consider rephrasing sentence “Anthocyanin intake from plant-derived food in human diet plays a protective role against coronary heart disease and with an improvement in sight”, particularly the last part (improvement in sight).
- Line 46: please change “anthocyanin biosynthesis pathway have been well elucidated” with “the anthocyanin biosynthesis pathway has been well elucidated”
- Lines 49-51 and so on: please revise gene and protein nomenclature. When referring to gene names should appear in uppercase and italics, while protein names should appear only in uppercase.
- Line 55 and lines 292-300: the terminology of structural/regulatory genes should appear in the introduction section and not in the discussion. On the other hand the authors should explain better why make a distinction between “structural” and “target” genes, because both are target genes of the TFs.
- Line 83: please change “were” with “have been”
- Line 86: please change “in combination” with “simultaneously” or “combined”

c) Regarding Results:
- Line 98: C1 abbreviation is already defined in line 64
- Line 106: figure 1 legend should include the definition of the gene abbreviations showed.
- Line 120-122: This is a discussion of the result obtained. Should be placed in the discussion section.
- Line 124 and Figure 2: please consider to include in figure 2 an indication of the gene constituvely expressed in each line (I think that would be easier for the reader). Also callus/plantlet age should be indicated in figure 2 legend.
- Line 154 and figure 5: Same suggestion as above. And indication of which gene is constituvely expressed as well as which probe is used in each assay (A,B,C or D) could make more easy to interpret the figure.
FISH analyses were performed also on the other transgenic lines? (AL31,32,41 and 44). These could be included in a supplementary figure.
- Lines 167-170: Figures 4 and 5 do not include any quantitative gene expression data that supports the sentence.
- Lines 173-175: Again the authors are interpreting the results obtained. Should be in the discussion section.
- Line 190: please introduce here the abbreviation of quantitative RT-PCR (qRTPCR) and use it accordingly in the rest of the manuscript (lines 256, 370, 492)
- Figures 6,7,9: qRT-PCR experiments: the authors should indicate which wheat sample has been used as a reference to calculate the transcript fold-change values. Also a statistical analysis must be performed and statistical differences should be indicated in the figures. Statistical methodology should also be detailed in material and methods.
- Figures 6 and 7: Bar Colors should be described in the figure legend. Also, I must recommend to the authors that they keep a common order of sample placement with Figures 4, 5, 6. Please check sample names/placements in figure 6, because I think that should be the same that are in figure 7.
- Figure 9 legend, line 268: please check definition of the tags used in the figure legend: AL-40G and AL41G are used for both control and light seeds. The number of biological replicates should be detailed
- Line 285, Figure 10: A statistical analysis should be performed. Also, error bars should be included. The number of biological replicates should be detailed.

d) Regarding Discussion:
- Line 301: the authors should consider rephrasing this sentence: “transform a gene into an organism” is not accurate term. The authors should try to point out more precisely the objective of this transformation: the constitutive expression of these two maize TFs in wheat.
- Line 312: please change “the dark pigmentation with “a dark pigmentation”
- Line 314-315: The interpretation that a differential accumulation of anthocyanin pigmentation between tissues in lines AL30-Al-31 reflects “the tissue-specific expression of R2R3-MYB type transcription factor in wheat” seems confusing for me. Are you referring to the wheat gene (TaR) or ZmR1? Please remember that ZmR1 is under the control of pUbi promoter and hence constituvely expressed. Also, there could be other control mechanisms downstream of gene expression that could affect to the activity of this TF, or the levels/activity of the proteins coded by the target genes of this TF (like the enzymes of the anthocyanin pathway) that could affect to the anthocyanin levels.
- Lines 327-331: The affirmation made by the authors should be backed up by qRT-PCR tissue expression studies.
- Lines 337-339: The affirmation made by the authors should be backed up by qRT-PCR expression studies.
- Line 353: please change “the Arabidopsis” with “Arabidopsis”
- Lines 356-357: “biosynthesis genes” is a term more accurate than “biosynthetic genes”
- Line 371: figure 8 do not show any qRT-PCR data

e) Regarding conclusion
- Line 402: please replace “combination expression of ” with “combined overexpression of”
- The authors should clarify how the transcription factors studied could be used as markers for wheat transformation

f) Regarding Materials and Methods :
- Line 413: please replace “vernalization” with “stratification”
- Vectors used should be referenced.
- Line 444-445: tri-parent hybridization method for wheat transformation should be referenced
- Line 477-485: FISH protocol should be referenced
- Line 498: 2deltadeltaCT method should be referenced
- TaADP gene definition should be included and referenced.

g) Regarding Author contribution:
Please clarify which is the difference between “wheat transformation” and  “obtaining stable transgenic plants by molecular test”

h) An Additional doubt: the effects of light exposure were also tested in the other transgenic lines?

Author Response

Dear Reviewer,

Thanks a lot for your thoughtful comments on our manuscript entitled “Overexpression of maize ZmC1 and ZmR transcription factors in wheat regulates anthocyanin biosynthesis in tissue-specific manner” (ijms-603105). We have carefully revised the manuscript according to your comments and suggestions. All the changes are highlighted with red/blue color in the modified manuscript. Our point-by-point responses to your comments are presented below.

a) The authors have generated transgenic lines where ZmC1 and ZmR are under the control of ubiquitin1 promoter…..not correct, please change with another terms like “simultaneous overexpression” .

Response: Thank you for your nice suggestion. We have edited the text according to the suggestion.

b) Regarding Introduction:

- Lines 36-38: Please consider rephrasing sentence “Anthocyanin intake from plant-derived food in human diet plays a protective role against coronary heart disease and with an improvement in sight”, particularly the last part (improvement in sight).

Response: Thank you for pointing out the mistake. We have rephrased the sentence in the revised manuscript.

- Line 46: please change “anthocyanin biosynthesis pathway have been well elucidated” with “the anthocyanin biosynthesis pathway has been well elucidated”

Response: Thank you for your kind suggestion. We have changed it in the revised manuscript.

- Lines 49-51 and so on: please revise gene and protein nomenclature. When referring to gene names should appear in uppercase and italics, while protein names should appear only in uppercase.

Response: Thanks for the valuable suggestion. We have modified the gene and protein nomenclature in the revised manuscript.

- Line 55 and lines 292-300: the terminology of structural/regulatory genes should appear in the introduction section and not in the discussion. On the other hand the authors should explain better why make a distinction between “structural” and “target” genes, because both are target genes of the TFs.

Response: Thank you for your constructive comments. We have modified and made clear the gene classification in the revised manuscript.

-Line 83: please change “were” with “have been”.

Response: Thanks a lot. We have changed it in the revised manuscript.
- Line 86: please change “in combination” with “simultaneously” or “combined”

Response: Thanks for the suggestion. We have changed it in the revised manuscript.

c) Regarding Results:

- Line 98: C1 abbreviation is already defined in line 64

Response: Thank you very much. We have edited the text.

- Line 106: figure 1 legend should include the definition of the gene abbreviations showed.

Response: Thanks for the suggestion. We have edited the legend of Figure 1 in the revised manuscript.

- Line 120-122: This is a discussion of the result obtained. Should be placed in the discussion section.

Response: Thank you for your suggestion. We have removed it into discussion section in the revised manuscript.

- Line 124 and Figure 2: please consider to include in figure 2 an indication of the gene constrictively expressed in each line (I think that would be easier for the reader). Also callus/plantlet age should be indicated in figure 2 legend.

Response: Thank you for the good suggestion. We have modified Figure 2 and its legend in the revised manuscript.

- Line 154 and figure 3: Same suggestion as above. And indication of which gene is constituvely expressed as well as which probe is used in each assay (A, B, C or D) could make more easy to interpret the figure. FISH analyses were performed also on the other transgenic lines? (AL31, 32, 41 and 44). These could be included in a supplementary figure.

Response: Thank you for your nice comment. We have modified Figure 3 in the revised manuscript.

- Lines 167-170: Figures 4 and 5 do not include any quantitative gene expression data that supports the sentence.

Response: Thank you very much. We have edited it in the revised manuscript.

- Lines 173-175: Again the authors are interpreting the results obtained. Should be in the discussion section.

Response: Thanks for the suggestion. We have edited it in the revised manuscript.

- Line 190: please introduce here the abbreviation of quantitative RT-PCR (qRTPCR) and use it accordingly in the rest of the manuscript (lines 256, 370, 492)

Response: Thank you for the comment. We have modified the text according to the suggestion, in the revised manuscript.

- Figures 6, 7, 9: qRT-PCR experiments: the authors should indicate which wheat sample has been used as a reference to calculate the transcript fold-change values. Also a statistical analysis must be performed and statistical differences should be indicated in the figures. Statistical methodology should also be detailed in material and methods.

Response: Thank you for the adequate suggestion. Wild-type Fielder and Jingdong18 samples were used as a reference to calculate the transcript fold-change values.

- Figures 6 and 7: Bar Colors should be described in the figure legend. Also, I must recommend to the authors that they keep a common order of sample placement with Figures 4, 5, 6. Please check sample names/placements in figure 6, because I think that should be the same that are in figure 7.

Response: Thanks. We have described the colored coloumn, have performed Tukey’s test and revised the figures in the revised manuscript.

- Figure 9 legend, line 268: please check definition of the tags used in the figure legend: AL-40G and AL41G are used for both control and light seeds. The number of biological replicates should be detailed

Response: Thank you for the correction. We have revised the legend of Figure 9 in the revised manuscript.

- Line 285, Figure 10: A statistical analysis should be performed. Also, error bars should be included. The number of biological replicates should be detailed.

Response: Thank you for your nice suggestion. We have performed the Tukey’s test analysis on the data and detailed information has been included in the revised manuscript. The number of biological replicates was also added in the text.

d) Regarding Discussion:

- Line 301: the authors should consider rephrasing this sentence: “transform a gene into an organism” is not accurate term. The authors should try to point out more precisely the objective of this transformation: the constitutive expression of these two maize TFs in wheat.

Response: Thanks a lot. We have edited the text according to the suggestion.

- Line 312: please change “the dark pigmentation with “a dark pigmentation”

Response: Thanks for the suggestion. We have changed the word.

- Line 314-315: The interpretation that a differential accumulation of anthocyanin pigmentation between tissues in lines AL30-Al-31 reflects “the tissue-specific expression of R2R3-MYB type transcription factor in wheat” seems confusing for me. Are you referring to the wheat gene (TaR) or ZmR1? Please remember that ZmR1 is under the control of pUbi promoter and hence constitutively expressed. Also, there could be other control mechanisms downstream of gene expression that could affect to the activity of this TF, or the levels/activity of the proteins coded by the target genes of this TF (like the enzymes of the anthocyanin pathway) that could affect to the anthocyanin levels.

Response: Thank you for your comment. Here we are referring both of the wheat and maize TF related genes because the constitutive overexpression of the maize TF genes also enhanced the expression of the wheat TF genes as well as the downstream structural genes involved in the anthocyanin biosynthesis in wheat. We have made modification in the text according to the suggestion.

- Lines 327-331: The affirmation made by the authors should be backed up by qRT-PCR tissue expression studies.

Response: Thank you for your comment. We have revised the text in the manuscript.

- Lines 337-339: The affirmation made by the authors should be backed up by qRT-PCR expression studies.

Response: Thanks again for this comment. We have edited the text in the revised manuscript.

- Line 353: please change “the Arabidopsis” with “Arabidopsis”

Response: Thank you for the correction. We have revised it.

- Lines 356-357: “biosynthesis genes” is a term more accurate than “biosynthetic genes”

Response: Thanks a lot for the suggestion. We have replaced the word in the revised manuscript.

- Line 371: figure 8 do not show any qRT-PCR data

Response: Thanks for the comment. We have revised it.

e) Regarding conclusion

- Line 402: please replace “combination expression of ” with “combined overexpression of”

Response: Thank you for your suggestion. We have replaced the word in the revised manuscript.

- The authors should clarify how the transcription factors studied could be used as markers for wheat transformation

Response: Thank you for this comment. The transcription factors regulate the anthocyanin synthesis in plants and anthocyanin coloration can be also treated as a visible morphological marker during the selection of other linked transgenes. We have included it in the revised manuscript.

f) Regarding Materials and Methods:

- Line 413: please replace “vernalization” with “stratification”

Response: We are so sorry that we cannot accept this suggestion because “vernalization” is the right word for winter wheat to enter reproductive growth period from vegetative growth stage after low temperature treatment of more than one month.

- Vectors used should be referenced.

Response: Thanks for this suggestion. We have referenced the vector pWMB006 by citing a publication in our laboratory (Wang et al., Plant Biotech J, 2017, 15: 614-623.). But the vector pWMB111 constructed in our laboratory has not been published yet, we just put its whole map in Figure S6c.

- Line 444-445: tri-parent hybridization method for wheat transformation should be referenced

Response: Thank you. We have cited the reference in the revised manuscript.

-Line 477-485: FISH protocol should be referenced.

Response: Thank you for your suggestion. Reference has been cited in the revised manuscript.

-Line 498: 2deltadeltaCT method should be referenced

Response: Thank you for your comment. We have cited the reference in the revised manuscript.

- TaADP gene definition should be included and referenced.

Response: Thanks for this mention. We have definite the TaADP gene for Triticum aestivum adenosine diphosphate ribosylation factor in the text and the abbreviation list, and also referenced it (Paolacci et al., BMC Molecular Biology, 2009, 10:11).

g) Regarding Author contribution: Please clarify which is the difference between “wheat transformation” and “obtaining stable transgenic plants by molecular test”

Response: Here,wheat transformation” means to obtain the candidate T0 transgenic plants by transferring the maize transcription factors ZmR and ZmC1 genes into wheat via tissue culture mediated with Agrobacterium, and “obtaining stable transgenic plants by molecular test” referrers to get homozygous transgenic plants without any segregation in next a few generation confirmed by FISH analysis and PCR. We have changed the description accordingly

h) An Additional doubt: the effects of light exposure were also tested in the other transgenic lines?

Response: Effects of light exposure were tested only in the ZmR transgenic lines.

Thanks again for your time in reviewing our manuscript.

Sincerely yours,

Bisma Riaz and Xingguo Ye

bisma.riaz75@gmail.com, yexingguo@caas.cn

Institute of Crop Sciences

Chinese Academy of Agricultural Sciences

Round 2

Reviewer 2 Report

The authors of the publication did not send edits.
They sent only promises that they would correct the text.
Until the text is amended, the article cannot be accepted for publication.

Author Response

Thanks for your comments on our manuscript entitled “Overexpression of maize ZmC1 and ZmR transcription factors in wheat regulates anthocyanin biosynthesis in a tissue-specific manner” (ijms-603105). We have carefully revised the manuscript according to the comments generated by you, the other reviewer and the editor. All the changes have been highlighted in the modified manuscript. Our point-by-point responses to the comments of reviewers and the editor are below.

Reviewer 2

The authors of the publication did not send edits. They sent only promises that they would correct the text. Until the text is amended, the article cannot be accepted for publication.

Response:

Thanks for this criticism. We had a language editing company revise this manuscript before submission. We have asked the editing company to amend the present revised manuscript in English again. The language company has provided us a certificate of English editing for this manuscript, which is submitted by an additional file in this window.

Thanks again for your time in reviewing our manuscript.

Sincerely yours,

Bisma Riaz and Xingguo Ye

bisma.riaz75@gmail.com, yexingguo@caas.cn

Institute of Crop Sciences

Chinese Academy of Agricultural Sciences

Reviewer 4 Report

The revised version of the manuscript has improved with the changes made, but there are still a number of issues that should be addressed:

a) Regarding abstract

-lines 11-12:

English grammar of the sentence “In this study, ZmC1 and ZmR were transformed into wheat one by one or together mediated with Agrobacterium for developing anthocyanin-enriched wheat germplasm” should be improved. Perhaps with something like “In this study, Agrobacterium-mediated transformation was used to generate plants that overexpress ZmC1 or ZmR or both, with the objective of developing anthocyanin-enriched wheat germplasm”

-lines 16-18: There are still not so much correct expressions like “expression only of ZMC1” or “expression only of ZmR”….again, I must insist that these must be changed to more correct expressions like "single overexpression"

-lines 22-23: The sentence “combined overexpression of ZmC1 and ZmR accumulated the highest pigment products” should be changed to something like “wheat plants with combined overexpression of ZmC1 and ZmR accumulated the highest quantity of these pigments products”

-lines 23-26: the sentence ” Moreover, up-regulated transcript levels of anthocyanin biosynthesis related genes in the treated developing seeds under light-exposed condition compared with the control developing seeds in ZmR transgenic lines led to a conclude that light increases the anthocyanin biosynthesis.” Could be improved, an example : “ Moreover, developing seeds overexpressing ZmR  exposed to light conditions show up-regulated transcript levels of anthocyanin biosynthesis related genes compared to dark exposure, thus suggesting  a role of  the light in the regulation of anthocyanin biosynthesis downstream the action of this transcription factor.”

b) Regarding Introduction:

-lines 37-38: I still feel that the sentence  “Anthocyanin intake from plant-derived food in human diet plays a protective role against coronary heart disease and also helps in an improvement in sight” could be improved to : ” Anthocyanin intake from plant-derived food in human diet plays a protective role against coronary heart disease and also helps in eyesight improvement”

 -lines 64-70:  again, please correct gene/protein nomenclature. In this paragraph you are describing roles and interaction between proteins, so the names should appear only in uppercase.

-Line 88: please change “to be an activator” with “to be activators” (I feel that you are refereeing to both TFs in the sentence)

-Line 89: ”please change “proteins” with “genes” (proteins do not have promoter regions or coding sequences!)

-Lines 91-93: the sentence “a trigenic cluster MbHF35 containing three genes HvMYB4H, HvMYC4H and HvF35H in conferring the grains with blue anthocyanin was identified in barley” could be improved to “a trigenic cluster MbHF35 containing three genes HvMYB4H, HvMYC4H and HvF35H, that confers blue anthocyanin to the grains, was identified in barley”

c) Regarding Results:

-lines 112-113: Again, be precise with gene and protein nomenclature:  C1 and R should not appear in italics.

-lines 119-120: there are abbreviations still absent in figure 1 legend: RB, LB and Bar

-lines 124-125: sentence “while the immature embryos transformed with pWMB196 in the experiment named NC-32 using non-purple wheat cultivar Fielder or pWMB198 in the experiment named NC-33 also using non-purple wheat cultivar Fielder didn’t show any purple color (Figure 2A)” could be improved to a  less repetitive sentence, like “while the immature embryos transformed with pWMB196 in the experiment named NC-32 or pWMB198 in the experiment named NC-33, both of them using non-purple wheat cultivar Fielder, didn’t show any purple color (Figure 2A)”

-line 177: please change “showed the tissue-specific anthocyanin accumulation” with “showed a tissue-specific anthocyanin accumulation”. Also, please change “In detailed” with “In detail”

-lines 181-185:  the authors are interpreting the results obtained. Should be in the discussion section. Also, change “accumulatation” with “accumulation”. And also:  you are talking about the proteins, not the genes, modify the nomenclature accordingly.

-lines 219-227: Figure 6, lines 219-227: Figure 7 and lines 285-289: Figure 9

Again my main concern is about how qRT-PCR experiments are calculated and showed:

Figure 6: the use of the untransformed wild-type lines wheat lines as a reference to calculate the transcript fold change values is a bad choice in the case of the evaluation of the expression of ZmC1 and ZmR lines, because you should not have any amplification in the PCR reaction in the untransformed wheat lines. You should have to use any of the transgenic lines that contain the gene in each case.  Please choose one and recalculate expressions. The line used for reference should be indicated in the figure legend

Figures 6 and 7: in the case of the evaluation of the expression of endogenous wheat genes, the choice of using the untransformed lines is correct. However, I do not see this reflected in the figure: taking these lines as the reference to calculate relative levels of mRNA means that the values obtained for these untransformed lines should be "1". Please revise and recalculate values. Also, an indication of the use of untransformed lines as reference should be indicated.

Figure 9: same concerns here, the authors should indicate which line or condition is used as reference to calculate the relative expression in the figure.

d) Regarding Discussion:

- Line 376: please change “the Arabidopsis” with “Arabidopsis”

Lines 379: I must insist,  “biosynthesis genes” is a term more accurate than “biosynthetic genes"

f) Regarding Materials and Methods:

Line 529: Please change “from the previous study” to “from a previous study”

Lines 533 and 555: : Please change “The differences among the transgenic wheat lines were separated by” to “The differences among the transgenic wheat lines were evaluated by”

Lines 534 and 556:  “Statistics 8.1” software should be referenced

Regarding supplementary material:

Table S1: The table could be simplified. Replication columns could be substituted by a single column of mean +- SD. Also, a statistical analysis should be performed.

Author Response

Thanks for your comments on our manuscript entitled “Overexpression of maize ZmC1 and ZmR transcription factors in wheat regulates anthocyanin biosynthesis in a tissue-specific manner” (ijms-603105). We have carefully revised the manuscript according to the comments generated by you, the other reviewer and the editor. All the changes have been highlighted in the modified manuscript. Our point-by-point responses to the comments of reviewers and the editor are below.

Reviewear 4:

a) Regarding abstract

-lines 11-12:

English grammar of the sentence “In this study, ZmC1 and ZmR were transformed into wheat one by one or together mediated with Agrobacterium for developing anthocyanin-enriched wheat germplasm” should be improved. Perhaps with something like “In this study, Agrobacterium-mediated transformation was used to generate plants that overexpress ZmC1 or ZmR or both, with the objective of developing anthocyanin-enriched wheat germplasm”.

Response: Thank you for your comment and suggestion. We have revised the text according to this suggestion.

-lines 16-18: There are still not so much correct expressions like “expression only of ZMC1” or “expression only of ZmR”….again, I must insist that these must be changed to more correct expressions like "single overexpression".

Response: Thanks a lot for your suggestion and comment. We have carefully modified the text in the revised manuscript.

-lines 22-23: The sentence “combined overexpression of ZmC1 and ZmR accumulated the highest pigment products” should be changed to something like “wheat plants with combined overexpression of ZmC1 and ZmR accumulated the highest quantity of these pigments products”.

Response: Thank you for your comment and suggestion. We have revised the text according to the suggestion.

-lines 23-26: the sentence ” Moreover, up-regulated transcript levels of anthocyanin biosynthesis related genes in the treated developing seeds under light-exposed condition compared with the control developing seeds in ZmR transgenic lines led to a conclude that light increases the anthocyanin biosynthesis.” Could be improved, an example : “ Moreover, developing seeds overexpressing ZmR  exposed to light conditions show up-regulated transcript levels of anthocyanin biosynthesis related genes compared to dark exposure, thus suggesting  a role of  the light in the regulation of anthocyanin biosynthesis downstream the action of this transcription factor.”

Response: Thanks a lot. We have modified the text according to the suggestion in the revised manuscript.

b) Regarding Introduction:

-lines 37-38: I still feel that the sentence  “Anthocyanin intake from plant-derived food in human diet plays a protective role against coronary heart disease and also helps in an improvement in sight” could be improved to : ” Anthocyanin intake from plant-derived food in human diet plays a protective role against coronary heart disease and also helps in eyesight improvement”.

Response: Thank you for your comment and suggestion. We have revised the text according to your suggestion in the revised manuscript.

 -lines 64-70:  again, please correct gene/protein nomenclature. In this paragraph you are describing roles and interaction between proteins, so the names should appear only in uppercase.

Response: Thank you for your comment. We have corrected them in the revised manuscript.

-Line 88: please change “to be an activator” with “to be activators” (I feel that you are refereeing to both TFs in the sentence).

Response: Thanks a lot for the comment. We have revised the text by adding word “activators”.

-Line 89: ”please change “proteins” with “genes” (proteins do not have promoter regions or coding sequences!).

Response: Thank you for your nice suggestion. We have changed the text by writing “genes” in the revised manuscript.

-Lines 91-93: the sentence “a trigenic cluster MbHF35 containing three genes HvMYB4H, HvMYC4H and HvF35H in conferring the grains with blue anthocyanin was identified in barley” could be improved to “a trigenic cluster MbHF35 containing three genes HvMYB4H, HvMYC4H and HvF35H, that confers blue anthocyanin to the grains, was identified in barley”.

Response: Thank you for this suggestion. We have modified the text in the revised manuscript using your edit and a suggested change from our editor.

c) Regarding Results:

-lines 112-113: Again, be precise with gene and protein nomenclature:  C1 and R should not appear in italics.

Response: Thank you for your suggestion. We have revised the text according to the suggestion.

-lines 119-120: there are abbreviations still absent in figure 1 legend: RB, LB and Bar.

Response: Thank you. We have included the abbreviations for RB, LB and Bar in the Figure 1 legend in the revised manuscript.

-lines 124-125: sentence “while the immature embryos transformed with pWMB196 in the experiment named NC-32 using non-purple wheat cultivar Fielder or pWMB198 in the experiment named NC-33 also using non-purple wheat cultivar Fielder didn’t show any purple color (Figure 2A)” could be improved to a  less repetitive sentence, like “while the immature embryos transformed with pWMB196 in the experiment named NC-32 or pWMB198 in the experiment named NC-33, both of them using non-purple wheat cultivar Fielder, didn’t show any purple color (Figure 2A)”.

Response: Thanks for the comment. We have modified the text according to the suggestion and a suggestion from our editor.

-line 177: please change “showed the tissue-specific anthocyanin accumulation” with “showed a tissue-specific anthocyanin accumulation”. Also, please change “In detailed” with “In detail”.

Response: Thank you for your suggestion. We have revised the text according to the suggestions and a suggestion from our editor.

-lines 181-185:  the authors are interpreting the results obtained. Should be in the discussion section. Also, change “accumulatation” with “accumulation”. And also:  you are talking about the proteins, not the genes, modify the nomenclature accordingly.

Response: Thank you for your comment. We have revised it.

-lines 219-227: Figure 6, lines 219-227: Figure 7 and lines 285-289: Figure 9

Again my main concern is about how qRT-PCR experiments are calculated and showed:

Figure 6: the use of the untransformed wild-type lines wheat lines as a reference to calculate the transcript fold change values is a bad choice in the case of the evaluation of the expression of ZmC1 and ZmR lines, because you should not have any amplification in the PCR reaction in the untransformed wheat lines. You should have to use any of the transgenic lines that contain the gene in each case.  Please choose one and recalculate expressions. The line used for reference should be indicated in the figure legend.

Response: Thank you for your suggestion. We have recalculated the expression values using transgenic line AL-45 as a reference sample and have also indicated it in the modified Figure 6A in the revised manuscript.

Figures 6 and 7: in the case of the evaluation of the expression of endogenous wheat genes, the choice of using the untransformed lines is correct. However, I do not see this reflected in the figure: taking these lines as the reference to calculate relative levels of mRNA means that the values obtained for these untransformed lines should be "1". Please revise and recalculate values. Also, an indication of the use of untransformed lines as reference should be indicated.

Response: Thanks for your comment. Expression values have been recalculated and wild-type Fielder was used as reference sample. This is also indicated in modified Figures 6B and 7 in the revised manuscript.

Figure 9: same concerns here, the authors should indicate which line or condition is used as reference to calculate the relative expression in the figure.

Response: Thank you. We have recalculated the expression values by using the control seeds of the AL-40 transgenic line as a reference sample and have also indicated it in the modified Figure 9 in the revised manuscript.

d) Regarding Discussion:

- Line 376: please change “the Arabidopsis” with “Arabidopsis”.

Response: Thanks. We have deleted the “the” in the revised manuscript.

Lines 379: I must insist,  “biosynthesis genes” is a term more accurate than “biosynthetic genes".

Response: Thank you. We have replaced the word “biosynthetic genes” with “biosynthesis genes” in the revised manuscript.

f) Regarding Materials and Methods:

Line 529: Please change “from the previous study” to “from a previous study”.

Response: Thank you for nice suggestion. We have changed it in the revised manuscript.

Lines 533 and 555: : Please change “The differences among the transgenic wheat lines were separated by” to “The differences among the transgenic wheat lines were evaluated by”.

Response: Thank you for your suggestion. We have changed it in the revised manuscript.

Lines 534 and 556:  “Statistix8.1” software should be referenced.

Response: Thank you. We have cited the reference for this program in the revised manuscript.

Regarding supplementary material:

Table S1: The table could be simplified. Replication columns could be substituted by a single column of mean +- SD. Also, a statistical analysis should be performed.

Response: Thank you for your suggestion. We have simplified the data and added the statistical results in the revised supplementary Table S1.

Thanks again for your time in reviewing our manuscript.

Sincerely yours,

Bisma Riaz and Xingguo Ye

bisma.riaz75@gmail.com, yexingguo@caas.cn

Institute of Crop Sciences

Chinese Academy of Agricultural Sciences

Round 3

Reviewer 2 Report

16 line - "ZmC1" - "ZmC1"

54 line - "chalcone synthase (CHS), chalcone isomerase (CHI), and flavanone 3-hydroxylase (F3H) dihydroflavonol 4-reductase (DFR) and anthocyanidin synthase (ANS)" - "chalcone synthase (CHS), chalcone isomerase (CHI), flavanone 3-hydroxylase (F3H), dihydroflavonol 4-reductase (DFR) and anthocyanidin synthase (ANS)"

85 line - "Tamyc1""TaMyc1" 

91 line - "TaPpm1 (purple pericarp-MYB 1) and TaPpb1 (purple pericarp-bHLH 1)" - "TaPpm1 (purple pericarp-MYB 1) and TaPpb1 (purple pericarp-bHLH 1)"

214, 365 lines - "TaMYC1" - "TaMyc1"

219 line - "TaC1 and TaR" - "TaPpm1 and TaPpb1"

279 line - "TaR" - "TaPpb1"

321 line - ; -:

321, 328 lines - "regulatory genes/TFs" - "regulatory genes"

Author Response

Dear reviewer,

Thank you very nuch for your reviewing again on our manuscript entitled “Overexpression of maize ZmC1 and ZmR transcription factors in wheat regulates anthocyanin biosynthesis in a tissue-specific manner” (ijms-603105). We have carefully revised the manuscript according to your comments. Our point-by-point responses to your suggestions are showed below.

16 line - "ZmC1" - "ZmC1"

Response: Thanks a lot for the correction. We have changed it accordingly.

54 line - "chalcone synthase (CHS), chalcone isomerase(CHI), and flavanone 3-hydroxylase (F3H) dihydroflavonol 4-reductase (DFR) and anthocyanidin synthase (ANS)" - "chalcone synthase(CHS), chalcone isomerase (CHI), flavanone 3-hydroxylase (F3H), dihydroflavonol 4-reductase (DFR) and anthocyanidin synthase (ANS)"

Response: These are the enzymes names. According to the normal requirements in most journals and the suggestions from the other reviewers, the names for proteins (enzymes) should be printed in uppercase.

85 line - "Tamyc1""TaMyc1"

Response: Thank you so much for your suggestion. We have modified this word in the revised manuscript.

91 line - "TaPpm1(purple pericarp-MYB 1) and TaPpb1(purple pericarp-bHLH 1)" - "TaPpm1(purple pericarp-MYB 1) and TaPpb1 (purple pericarp-bHLH 1)"

Response: Thank you. We have modified them in the revised manuscript.

214, 365 lines - "TaMYC1" - "TaMyc1"

Response: Thank you for your comment. We have corrected it.

219 line - "TaC1and TaR" - "TaPpm1and TaPpb1"

Response: Thanks a lot. We have written the specific gene names in the revised manuscript.

279 line - "TaR" - "TaPpb1"

Response: Thank you. We have modified it.

321 line - ; -:

Response: Thanks for your comment. We have corrected the punctuation.

321, 328 lines - "regulatory genes/TFs" - "regulatory genes"

Response: Thank you. We have revised it.

Thanks again for your time in reviewing our manuscript.

Sincerely yours,

Bisma Riaz and Xingguo Ye

bisma.riaz75@gmail.com, yexingguo@caas.cn

Institute of Crop Sciences

Chinese Academy of Agricultural Sciences

Reviewer 4 Report

Dear Authors

I am happy to see that the main issues pointed previously have been addressed. Still, some minor issues remain:

a) Lines 229-234: the meaning of the start of this paragraph is a bit confusing: the level of a transcript encoding an enzyme do not have any involvement in the pathway were this protein performs its function. These sentences need some rewriting. A suggestion:
“A selection of wheat endogenous structural genes encoding enzymes involved in anthocyanin biosynthesis pathways including TaCHS (unigene c54121_g1_i2), TaCHI (unigene c49033_g1_i1), TaF3H (unigene 231 c57117_g3_i1), TaF3'H (unigene c55981_g1_i1), TaDFR (unigene c58412_g2_i1), and TaANS (unigene 232 c57117_g3_i5) was performed (Supplementary Figure S5). Their expression levels were evaluated in the leaves of the three types of stable transgenic lines as well as both wild-type plants”

b) Line 257: if a change is not statistically significant means that there is no change, so the sentence “the expression level of TaCHI was up-regulated (but not significantly) in the ZmC1 containing lines” is incorrect, please rewrite.

c) Lines 244-246: The sentence “However, the transcript levels of TaDFR and TaANS were not significantly upregulated in ZmC1 containing transgenic lines (AL-30, AL-31, and AL-32), and ZmR containing transgenic lines (AL-40 and AL-41) (Figure 7).” is not accurate, because the expression levels of TaDFR have significant differences in the case of AL30,31,32 lines compared to the wild type lines. Also the expression levels of TaANS in ZmC1-containing transgenic lines and ZmR-containing transgenic lines have significant differences compared to wild type lines. Please rephrase with a more accurate description of your results.

d) Lines 280-281: I suggest changing the sentence to “In this experiment, the untreated AL-40 seeds were used as the reference sample to calculate the relative expression levels”

e) Line 286: please “ZMR containing transgenic wheat plants” with “ZMR-containing transgenic wheat plants”

f) Figure 9: As a suggestion, I think that the figure will be easier to follow by the reader if different colors were used to distinguish the light-treated seeds from the untreated ones. Also, I think that “material used as a reference” could be changed with “sample used as a reference” in the figure 9 legend

g) Line 306: The numbers regarding anthocyanin contents indicated are different from the numbers appearing in supplementary table S1. Please correct them. Also, these numbers should be expressed with their SD.

h) Line 385: like “biosynthetic genes”, the expression “enzymatic genes” should be changed. Perhaps the sentence could be rephrased to “In Arabidopsis, genes encoding enzymes of the anthocyanin biosynthesis pathway are grouped into two classes”

i) Line 433: please change “ZmC1 and ZmR” with “ZmC1 or ZmR”, because you are referring to the single-gene overexpressing lines

j) Line 462: I think that the sentence could be modified with “resulted in increased anthocyanin content in a tissue specific manner”

k) Supplementary material document should include the title of the paper and the names of the authors. Also, Table S1 should include the statistical analysis showed in figure 10.

Author Response

Dear reviewer,

Thank you very nuch for your reviewing again on our manuscript entitled “Overexpression of maize ZmC1 and ZmR transcription factors in wheat regulates anthocyanin biosynthesis in a tissue-specific manner” (ijms-603105). We have carefully revised the manuscript according to your comments. Our point-by-point responses to your suggestions are showed below.

a) Lines 229-234: the meaning of the start of this paragraph is a bit confusing: the level of a transcript encoding an enzyme do not have any involvement in the pathway were this protein performs its function. These sentences need some rewriting. A suggestion: 
“A selection of wheat endogenous structural genes encoding enzymes involved in anthocyanin biosynthesis pathways including TaCHS (unigene c54121_g1_i2), TaCHI (unigene c49033_g1_i1), TaF3H (unigene 231 c57117_g3_i1), TaF3'H (unigene c55981_g1_i1), TaDFR (unigene c58412_g2_i1), and TaANS (unigene 232 c57117_g3_i5) was performed (Supplementary Figure S5). Their expression levels were evaluated in the leaves of the three types of stable transgenic lines as well as both wild-type plants”.

Response: Thank you for your nice suggestion. We have modified the sentences according to the suggestion.

b) Line 257: if a change is not statistically significant means that there is no change, so the sentence “the expression level of TaCHI was up-regulated (but not significantly) in the ZmC1 containing lines” is incorrect, please rewrite.

Response: Thanks a lot for your comment. We have corrected them in the revised manuscript.

c) Lines 244-246: The sentence “However, the transcript levels of TaDFR and TaANS were not significantly upregulated in ZmC1 containing transgenic lines (AL-30, AL-31, and AL-32), and ZmR containing transgenic lines (AL-40 and AL-41) (Figure 7).” is not accurate, because the expression levels of TaDFR have significant differences in the case of AL30,31,32 lines compared to the wild type lines. Also the expression levels of TaANS in ZmC1-containing transgenic lines and ZmR-containing transgenic lines have significant differences compared to wild type lines. Please rephrase with a more accurate description of your results.

Response: Thanks for your comment. We have corrected the description of the results in the revised manuscript.

d) Lines 280-281: I suggest changing the sentence to “In this experiment, the untreated AL-40 seeds were used as the reference sample to calculate the relative expression levels”.

Response: Thank you for your nice suggestion. We have changed the sentence according to the suggestion.

e) Line 286: please “ZMR containing transgenic wheat plants” with “ZMR-containing transgenic wheat plants”.

Response: Thanks a lot. We have revised it according to the suggestion.

f) Figure 9: As a suggestion, I think that the figure will be easier to follow by the reader if different colors were used to distinguish the light-treated seeds from the untreated ones. Also, I think that “material used as a reference” could be changed with “sample used as a reference” in the figure 9 legend.

Response: Thank you. We have changed the figure according to suggestion and also revised the legend of the figure.

g) Line 306: The numbers regarding anthocyanin contents indicated are different from the numbers appearing in supplementary table S1. Please correct them. Also, these numbers should be expressed with their SD.

Response: Thanks for pointing out the shortage. We have added SD in these data.

h) Line 385: like “biosynthetic genes”, the expression “enzymatic genes” should be changed. Perhaps the sentence could be rephrased to “In Arabidopsis, genes encoding enzymes of the anthocyanin biosynthesis pathway are grouped into two classes”.

Response: Thanks. We have modified the sentence according to the suggestion.

i) Line 433: please change “ZmC1 and ZmR” with “ZmC1 or ZmR”, because you are referring to the single-gene overexpressing lines.

Response: Thank you for the suggestion. We have changed it in the revised manuscript.

j) Line 462: I think that the sentence could be modified with “resulted in increased anthocyanin content in a tissue specific manner”.

Response: Thank you. We have modified it.

k) Supplementary material document should include the title of the paper and the names of the authors. Also, Table S1 should include the statistical analysis showed in figure 10.

Response: Thank you for your suggestion. We have included the manuscript title and authors details in the supplementary material document; and also showed the statistical analysis in the Table S1.

Thanks again for your time in reviewing our manuscript.

Sincerely yours,

Bisma Riaz and Xingguo Ye

bisma.riaz75@gmail.com, yexingguo@caas.cn

Institute of Crop Sciences

Chinese Academy of Agricultural Sciences